# Kinetochore component function in *C. elegans* oocytes revealed by 4D tracking of holocentric chromosomes

Laras Pitayu-Nugroho[1,4], Mélanie Aubry[1,4], Kimberley Laband [1], Hélène Geoffroy[1], Thadshagine Ganeswaran[1], Audi Primadhanty [2], Julie C. Canman[3] & Julien Dumont [1] ✉

During cell division, chromosome congression to the spindle center, their orientation along the spindle long axis and alignment at the metaphase plate depend on interactions between spindle microtubules and kinetochores, and are pre-requisite for chromosome bi-orientation and accurate segregation. How these successive phases are controlled during oocyte meiosis remains elusive. Here we provide 4D live imaging during the first meiotic division in *C. elegans* oocytes with wild-type or disrupted kinetochore protein function. We show that, unlike in monocentric organisms, holocentric chromosome bi-orientation is not strictly required for accurate chromosome segregation. Instead, we propose a model in which initial kinetochore-localized BHC module (comprised of BUB-1[Bub1], HCP-1/2[CENP-F] and CLS-2[CLASP])-dependent pushing acts redundantly with Ndc80 complex-mediated pulling for accurate chromosome segregation in meiosis. In absence of both mechanisms, homologous chromosomes tend to co-segregate in anaphase, especially when initially mis-oriented. Our results highlight how different kinetochore components cooperate to promote accurate holocentric chromosome segregation in oocytes of *C. elegans*.

In female gametes, accurate segregation of homologous chromosomes and sister chromatids, during the first and second meiotic divisions respectively, is essential to avoid generating aneuploid eggs[1]. Fertilization of aneuploid oocytes is a major cause of infertility in humans, as it usually leads to spontaneous abortion or, if survived, leads to severe developmental defects[2–5]. To avoid this fate, chromosomes must interact properly with spindle microtubules in order to orient along the spindle axis, align at the metaphase plate and segregate in two equal complements during anaphase. The capacity of chromosomes to interact with microtubules is provided by the multiprotein kinetochores, which contain multiple microtubule-associated activities[6] (Fig. 1a). Among those, the KMN (Knl1/Mis12/Ndc80) network provides the core load-bearing activity essential for accurate

chromosome segregation in most species and cell types[7]. Within the KMN network, the tetrameric Ndc80 complex, together with its downstream partner SKA complex, can form end-on attachments to track depolymerizing kinetochore microtubules and generate pulling forces on chromosomes[7–18]. The Knl1 protein and the Mis12 complex are essential for proper kinetochore recruitment of the Ndc80 and SKA complexes[19–22]. Also downstream of Knl1 and Mis12, the Rod-Zw10-Zwilch (RZZ)-Spindly kinetochore module recruits dynein-dynactin motors to kinetochores, which mediate initial lateral capture of microtubules, ensuring correct kinetochore orientation and facilitating chromosome bi-orientation[23–28]. In *C. elegans*, a third kinetochore module, hereafter termed the BHC module, recruited downstream of KNL-1 is comprised of the kinase BUB-1, the two CENP-F-like proteins

[1]Université Paris Cité, CNRS, Institut Jacques Monod, F-75013 Paris, France. [2]Universitat Politècnica de Catalunya, 08028 Barcelona, Spain. [3]Columbia University Irving Medical Center; Department of Pathology and Cell Biology, New York, NY 10032, USA. [4]These authors contributed equally: Laras Pitayu-Nugroho, Mélanie Aubry. ✉e-mail: Julien.dumont@ijm.fr

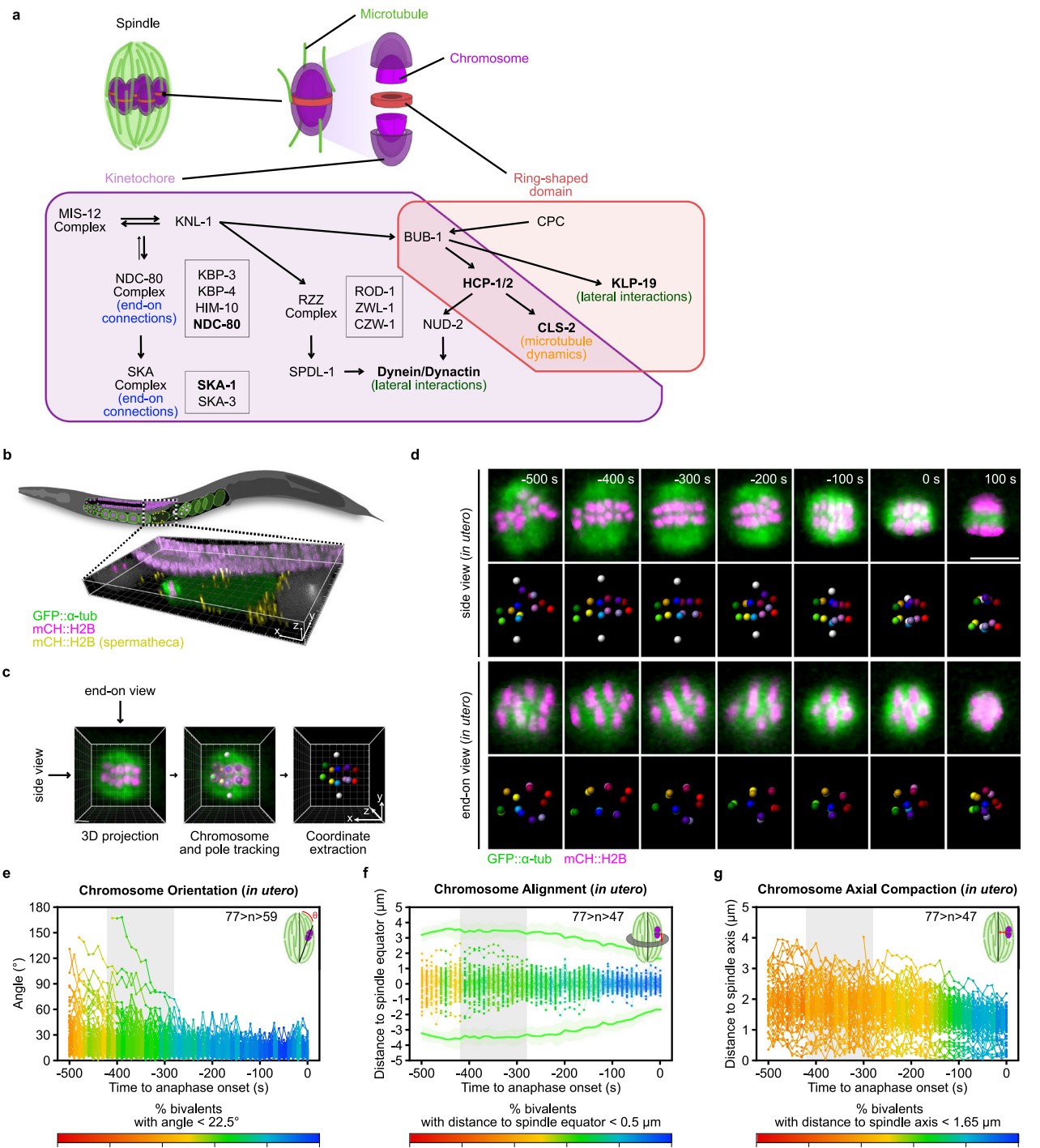

**Fig. 1 | Complete *in utero* 3D tracking of holocentric chromosomes in *C. elegans* oocytes. a** Top, Schematics of the *C. elegans* oocyte meiotic spindle with aligned chromosomes (left) and of an individual chromosome showing the meiotic cup-like kinetochores and ring-shaped domains (right). Bottom, Schematic of proteins and sub-complexes recruited at kinetochores (magenta) and ring-shaped domains (red). Proteins in bold display direct interaction with microtubules. **b** Schematic of *C. elegans* and its reproductive system. The inlet is centered on the anterior spermatheca, where fertilization occurs, and corresponds to the bottom 3D projection of a metaphase I fertilized oocyte expressing GFP::TBA-2^α-tubulin (green) and mCherry::HIS-11^H2B (magenta, germline chromatin; yellow, sperm chromatin after manual segmentation) passing through the spermatheca. Grid scale bar, 5 μm. **c** Schematic of the three-step process used to perform 3D tracking of chromosomes (colored spots) and spindle poles (white spots) over time. Grid scale bar, 1 μm. **d** Representative time-lapse images, centered on the meiotic spindle and chromosomes, of a GFP::TBA-2^α-tubulin (green) and mCherry::HIS-11^H2B (magenta)

-expressing oocyte undergoing the first meiotic division. Side projection of the spindle and chromosomes is shown at the top. End-on projection is at the bottom. Time relative to anaphase onset is indicated at the top right corner of each panel. Scale bar, 5 μm. **e–g** Plots of individual homologous chromosome pair angles relative to the spindle long axis (**e**), or of the distance between individual homologous chromosome pairs and the spindle equator, the mean distance between the spindle poles and equator are plotted as green curves with SD (**f**), or of the distance between individual homologous chromosome pairs and the spindle long axis (**g**), during the 500 s preceding anaphase I onset in oocytes. Each dot corresponds to an individual homologous chromosome pair and is color-coded as indicated at the bottom of each graph. Gray boxes highlight the SD of the spindle bipolarization timing centered on the mean timing. Schematics of the measured parameter and sample size (*n* pairs of homologous chromosomes) are at the top right corner of each graph.

HCP-1/2 and the CLASP-family member CLS-2[29–32]. We recently showed that BHC module components synergistically stabilize microtubules by promoting rescue and pause, while preventing catastrophe, which is in line with the function of CLASP-family proteins in other systems[33–38]. BHC module kinetochore localization depends on BUB-1 binding to a series of MELT repeats located N-terminally in KNL-1[20,33,39–44]. As BHC module components are also present at ring domains located between chromosomes and along the spindle, the specific function of the BHC module at kinetochores is unclear[45–47].

Unlike monocentric species, which display point-shaped kinetochores, holocentric organisms have diffuse kinetochores that run along the length of sister chromatids[48–50]. In oocytes of the holocentric nematode *Caenorhabditis elegans*, kinetochores form cup-shaped structures that cover the surface of the six oval chromosomes in meiosis I and II[51,52]. Although, most outer kinetochore components, including the KMN network, are shared between monocentric and holocentric species, and are almost all found at the cup-shaped kinetochores of *C. elegans* oocytes, the exact function of kinetochore proteins in meiosis is controversial[45]. Further, we and others found that in *C. elegans* oocytes anaphase chromosome separation is primarily driven by a kinetochore-independent atypical pushing mechanism involving central spindle microtubules assembled in a CLS-2-dependent manner[45,46,53,54]. Two studies reported that only lateral interactions between chromosomes and microtubules form during prometaphase/metaphase[55,56]. However, a recent study reported that end-on kinetochore-microtubule attachments are predominant in prometaphase/metaphase and are essential for chromosome separation in anaphase[57].

Here we have addressed this issue, using 4D simultaneous two-color confocal fluorescence microscopy at high spatiotemporal resolution in live *C. elegans* oocytes. We successfully tracked all six bivalent chromosomes and the meiotic spindle with unprecedented spatial and temporal resolution during the first meiotic division in oocytes with wild-type or disrupted kinetochore protein function. These datasets allowed a systematic and comprehensive quantitative analysis of the kinetochore contribution to chromosome movements during meiotic prometaphase/metaphase. We show that chromosome orientation along the spindle long axis is tightly coupled to spindle bipolarization, but does not involve appreciable chromosome oscillations along the spindle. By systematically depleting each kinetochore sub-complex alone or in combination, we assigned specific functions (or lack thereof) to lateral and end-on interactions between chromosomes and spindle microtubules. We found that depletion of the kinetochore scaffold protein KNL-1, which completely prevents kinetochore assembly, led to severe chromosome orientation and alignment defects in prometaphase/metaphase, followed by highly inaccurate chromosome segregation in anaphase. In an effort to recapitulate this severe meiotic kinetochore-null phenotype by perturbing microtubule-associated activities, we found that KLP-19 and dynein/dynactin-mediated lateral interactions promoted timely chromosome orientation and congression in mid-prometaphase. In contrast, Ndc80 complex-dependent end-on attachments were specifically required for chromosome alignment throughout prometaphase/metaphase. However and despite their function prior to anaphase, neither lateral interactions, nor end-on attachments, were essential on their own or in combination for accurate chromosome segregation in meiosis. We also found that kinetochore-targeted BHC module (BUB-1/HCP-1/2[CENP-F]/CLS-2[CLASP]) participated in efficient assembly of central spindle microtubules in anaphase, but did not play a significant role on its own in chromosome orientation or congression prior to anaphase onset. Unexpectedly, the meiotic kinetochore-null phenotype, obtained following KNL-1 depletion, could only be recapitulated by simultaneously depleting the Ndc80 complex and preventing BHC module kinetochore targeting. We propose a model in which initial BHC

module-dependent pushing acts redundantly with Ndc80 complex-mediated pulling for accurate chromosome segregation in meiosis. In absence of both mechanisms, homologous chromosomes have a higher tendency to co-segregate in anaphase, especially when initially mis-oriented relative to the spindle long axis. These results highlight how distinct roles for different kinetochore components cooperate to promote accurate chromosome segregation in meiosis.

## Results

### Full 4D tracking of chromosomes and spindle poles during meiosis I

To analyze chromosome movements relative to the spindle axes with high spatiotemporal accuracy during the first meiotic division in live *C. elegans* oocytes, we acquired four-dimensional (4D) *in utero* datasets on anaesthetized *C. elegans* adult worms stably expressing mCherry-H2B and EGFP-αTubulin during ovulation (Fig. 1b). To avoid blurring of the 3D stacks due to passage of the oocyte through the worm spermatheca during the first meiotic division, accompanied by fast movements of the spindle and chromosomes within the oocyte, we acquired the two fluorescent channels at each Z position simultaneously on two cameras. We combined the simultaneous two-channel acquisition with fast Z-scanning using a piezo-controlled motorized stage and a super-fast stream acquisition mode (10 ms per acquisition) to acquire 9 μm-thick Z-stacks (30×0.3 μm) every 10 s throughout meiosis I and II. Live imaging did not appreciably perturb cell division as evidenced by successful first polar body extrusion in 100% of oocytes. These 4D datasets were processed to semi-automatically segment and track all chromosomes and the spindle poles (Fig. 1c, d and Supplementary Movie 1). The resulting individual chromosome and spindle pole tracks were interactively validated and manually corrected as needed.

To understand the mechanisms that govern chromosome movements inside the spindle, we first analyzed the kinetics of chromosome behavior at high spatiotemporal resolution. Specifically, we analyzed three parameters: (1) chromosome orientation relative to the spindle long axis, (2) chromosome congression to the spindle equator, and (3) chromosome axial compaction along the spindle short axis. We began our analysis in late prometaphase starting ~500 s before anaphase I onset (Fig. 1e–g), as we could not accurately analyze these parameters earlier in prometaphase due to spindle isotropicity and lack of a clear spindle axis[58,59]. The average maximum chromosome-spindle angle at anaphase onset was observed to be 22.5°, below which chromosomes were considered to be properly oriented (Fig. 1e and Supplementary Fig. 1a). To evaluate chromosome orientation over time, we color-coded the proportion of chromosomes with an angle below this threshold value. This analysis revealed that chromosome orientation correlated with spindle bipolarization (gray rectangle) when >50% of chromosomes were properly oriented. We only observed a few chromosomes ($n = 4/77$) that reverted orientation (angles > 90°) during meiosis I. This is in stark contrast with the behavior of monocentric chromosomes in mouse oocytes, which revert orientation frequently through successive cycles of bi-orientation and detachment[60]. In contrast, our results suggested coupling between spindle bipolarization and chromosome orientation in *C. elegans* oocytes.

Using a similar approach, we analyzed chromosome congression and lateral compaction. The average maximum distance between chromosomes and the spindle equator in metaphase, which we considered as the threshold value for correct congression, was 0.5 μm (Fig. 1f and Supplementary Fig. 1a). Chromosomes did not oscillate along the spindle during meiosis I, but rather seemed constrained at the spindle center and progressively reached the metaphase plate (Supplementary Fig. 1b). This is again different from the oscillatory behavior of monocentric mouse oocytes during their alignment on the metaphase plate[60]. We observed a progressive lateral compaction of the metaphase plate leading to increased proximity between

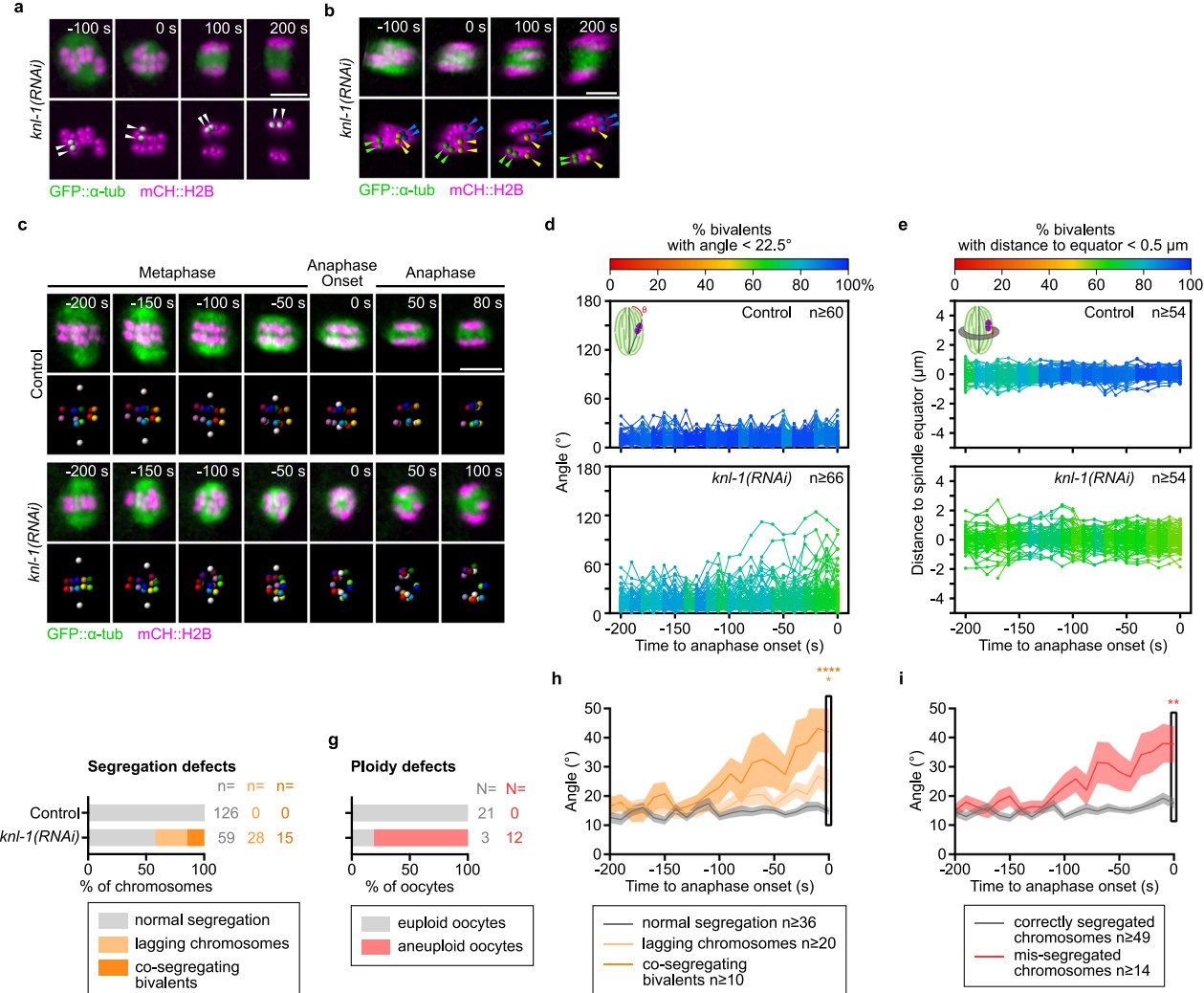

**Fig. 2 | The meiotic kinetochore-null phenotype is associated with homologous chromosome mis-orientation, mis-alignment and a high rate of oocyte aneuploidy. a–c** Representative time-lapse images of GFP::TBA-2^{α-tubulin} (green) and mCherry::HIS-11^{H2B} (magenta) -expressing oocytes during end of metaphase and anaphase in the indicated conditions. White arrowheads in (**a**) highlight two homologous chromosomes of a pair undergoing co-segregation. Green, yellow and blue arrowheads in (**b**) show the co-segregating homologous chromosomes of a pair, lagging chromosome of a pair that eventually segregates correctly, and lagging chromosome of a pair that ends-up mis-segregated respectively. Time relative to anaphase onset is indicated at the top right corner of each panel. Scale bars, 5 μm. **d, e** Plots of individual homologous chromosome pair angles relative to the spindle long axis (**d**) or of the distance between individual homologous chromosome pairs and the spindle equator (**e**) during 200 s preceding anaphase I onset in

oocytes in the indicated conditions. The color-code is indicated at the top of each graph. Schematics of the measured parameter and sample size (*n* pairs of homologous chromosomes) are at the top left and right corners respectively of each graph. **f, g** Quantification of normally segregating, lagging and co-segregating chromosomes during anaphase (**f**), and euploid or aneuploid oocytes at the end of anaphase (**g**) in indicated conditions. Sample sizes (*n* chromosomes, N oocytes) are at the right of each graph. **h, i** Mean angle of homologous chromosome pairs relative to the spindle long axis over time for (**h**) normally segregating (gray), lagging (light orange) and co-segregating (dark orange), or (**i**) correctly segregated (gray) and mis-segregated (red) chromosomes in oocytes treated with *knl-1(RNAi)*. Dark lines, mean; lighter bands, SEM. Sample sizes (*n* chromosomes) are in the key. **h** one-way ANOVA with Tukey's multiple comparison; **i** *t*-test with Welch's correction, two-sided: n.s., $P \geq 0.05$; **$P < 0.01$; ***$P < 0.001$; and ****$P < 0.0001$.

chromosomes, along the spindle short axis. This axial compaction seemed to passively follow spindle compaction with no apparent coupling with spindle bipolarity or chromosome alignment (Fig. 1d, g). Together, our results show that holocentric chromosome dynamics in *C. elegans* oocytes is drastically different from that of monocentric meiotic chromosomes.

### Kinetochores are essential for chromosome orientation, congression and accurate segregation, but not physical separation of chromosomes during meiosis I

To gain functional insights into the mechanisms governing chromosome movements during prometaphase/metaphase I, we next analyzed the effect of depleting the kinetochore scaffold protein KNL-1 by RNAi[19,45] (Fig. 1a). We noticed that the chromosome segregation

phenotype of KNL-1-depleted oocytes was aggravated when filming *in utero* (even with low Z sampling, Supplementary Fig. 1c–e) as compared to filming oocytes *ex utero* (Fig. 2a–c, see discussion). We thus decided to generate 4D datasets of *ex utero*-filmed oocytes, which is compatible with high-throughput analyses yet restricted to the last 200 s preceding anaphase onset (Fig. 2c and Supplementary Movie 2). Using classical low spatiotemporal resolution imaging, we and others have previously shown that KNL-1 is essential for kinetochore assembly and accurate chromosome segregation, but not for chromosome anaphase physical separation. Our quantitative high-resolution 4D analysis confirmed these previous findings, but allowed discriminating co-segregating bivalents that were otherwise indistinguishable in commonly used 2D projections (Fig. 2a). Multiple evident chromosome mis-segregation events were observed in most KNL-1-depleted

oocytes, coupled to severe orientation (62.7% properly oriented chromosomes *vs* 92.1% in controls at anaphase onset) and congression defects (58.3% properly congressed chromosomes *vs* 95.2% in controls at anaphase onset) (Fig. 2b–e and Supplementary Movie 3). While congression seemed equally affected over time, chromosome orientation worsened as oocytes progressed toward metaphase I in absence of KNL-1 (80.3% properly oriented at −200 s *vs* 62.7% properly oriented at anaphase onset). We observed three distinct categories of chromosome segregation defects in KNL-1-depleted oocytes: (1) co-segregating bivalents, (2) mis-segregating lagging chromosomes, and (3) lagging chromosomes that eventually segregated properly (Fig. 2b, f and Supplementary Movie 3). The later did not induce oocyte aneuploidy, but co-segregating bivalents combined with mis-segregating lagging chromosomes accounted for the 80% aneuploid oocytes observed following KNL-1 depletion (Fig. 2g). By comparing the fate of chromosomes in anaphase to their orientation in prometaphase/metaphase over time, we observed that chromosomes displaying higher angles were more likely to become co-segregating bivalents (Fig. 2h). Furthermore, we found a strong correlation between a high angle at anaphase onset and the propensity to end-up mis-segregated (Fig. 2i). These results suggest that KNL-1 is essential for accurate chromosome segregation by controlling chromosome orientation prior to anaphase onset, but not for the physical chromosome separation in anaphase.

## Kinetochore dynein and KLP-19 act redundantly to promote initial chromosome orientation and congression

To determine which kinetochore-microtubule binding activities control accurate chromosome segregation, we analyzed the contribution of the main chromosomal components involved in contacts with microtubules (Fig. 1a). We included all kinetochore components with microtubule-associated activities downstream of KNL-1, and the chromokinesin KLP-19, which localizes independently of KNL-1 at ring-shaped domains located between homologous chromosomes and sister chromatids in meiosis I and II respectively[45,61–63].

Chromosome congression relies primarily on forces exerted by microtubule motors located on chromosomes arms (i.e., chromokinesins) and at kinetochores (CENP-E and dynein/dynactin)[23–28,64,65]. *C. elegans* lack a CENP-E ortholog. Thus, during mitosis, dynein/dynactin is the only kinetochore motor and is recruited to the kinetochores by the RZZ-Spindly (SPDL-1 in *C. elegans*) complex and stabilized at the kinetochores by the NudE/L ortholog NUD-2[24,66–68]. We first determined that this was also the case during oocyte meiosis. In line with a previous report, endogenously tagged DHC-1::mNG (the *C. elegans* dynein heavy chain) was undetectable at meiotic kinetochores upon depletion of the RZZ subunit ZWL-1[57] (Supplementary Fig. 2a, b). We obtained identical results upon depletion of NUD-2 by RNAi. However, neither depletion of ZWL-1, nor deletion of *nud-2*, nor a combination of both led to persistent and/or significant chromosome orientation, congression, or segregation defects, indicating that kinetochore-localized dynein/dynactin is not essential for accurate chromosome segregation in *C. elegans* oocytes (Supplementary Fig. 2c–g and Supplementary Fig. 3a, b). We next focused on the kinesin-4 family member KLP-19 (ortholog of the mammalian chromokinesin Kif4)[61]. During congression, chromokinesins generate pushing forces on chromosomes, termed polar ejection forces, and directed away from spindle poles[65]. In mitosis, KLP-19-mediated polar ejection forces orient holocentric kinetochores to prevent merotelic attachments, which otherwise lead to anaphase chromosome bridges and chromosome missegregation[61]. The exact function of KLP-19 in oocytes is still unclear. An initial report suggested that KLP-19 is essential for chromosome alignment[62]. While a second study found a more minor role of KLP-19, essential to maintain chromosomes aligned at the spindle center only when oocytes are artificially arrested in metaphase I, but not in normally progressing oocytes[45]. To conclusively address this

issue, we applied our 4D microscopy and quantitative tracking approach to KLP-19-depleted oocytes (Supplementary Fig. 2e). In line with the conclusions from the second study, we did not detect any significant chromosome orientation, congression or segregation defects after KLP-19 depletion (Supplementary Fig. 2c–g and Supplementary Fig. 3a, b). The lack of apparent phenotype upon loss of kinetochore dynein/dynactin or KLP-19 suggested that they might act redundantly to promote chromosome orientation and/or congression. To test this idea, we analysed oocytes simultaneously depleted of KLP-19 and ZWL-1 (Fig. 3a, second panel). Consistent with our hypothesis, in this condition we observed an increase in the rates of misoriented (25% misoriented chromosomes *vs* 5% in controls 200 s before anaphase onset) and misaligned chromosomes (42,6% misaligned chromosomes *vs* 30% in controls 200 s before anaphase onset) in prometaphase (Fig. 3b–e). However, chromosome orientation and alignment improved progressively over time following co-depletion of KLP-19 and ZWL-1, so that they reached levels comparable to control oocytes -100 s before anaphase onset (86,5% properly oriented chromosomes *vs* 94,7% in controls; 83,3% properly congressed chromosomes *vs* 91,2% in controls 70 s before anaphase onset) and overall chromosome segregation defects were only rarely observed (Fig. 3d–g). Overall, our results indicate that KLP-19-mediated polar ejection forces, combined with kinetochore-localized dynein/dynactin activity, participate in timely establishment of chromosome orientation and congression, but are not strictly required for accurate chromosome segregation in anaphase.

## The Ndc80 complex promotes efficient chromosome congression but is not essential for chromosome orientation, physical separation, or accurate segregation

The progressive improvement of chromosome orientation and congression in oocytes co-depleted of KLP-19 and ZWL-1 suggested that an alternative mechanism, activated before anaphase onset, can compensate for the lack of lateral interactions to promote formation of a tight metaphase plate. In mitosis, following initial motor- or kinetochore-mediated lateral interactions between chromosomes and spindle microtubules, kinetochores form end-on attachments to microtubules mediated by the conserved Ndc80 complex[7–11]. In *C. elegans* oocytes, the role of the four-subunit Ndc80 complex is controversial, with one report, based on electron tomography of fixed oocytes, supporting a lack of end-on attachments during metaphase, while another recent study, using *in utero* live microscopy, concluded on the essential function of the Ndc80 complex for chromosome segregation[56,57]. To clarify this issue and ensure full disruption of the Ndc80 complex, we combined auxin-induced degradation in a strain carrying an auxin-inducible degron (AID) added on the C-terminus of the endogenous Ndc80 complex subunit *him-10*, combined with RNAi-mediated depletion of the remaining HIM-10 protein[69] (Fig. 3a, third panel). In contrast to the inhibition of lateral interactions, preventing end-on attachments through HIM-10 depletion did not induce any detectable chromosome orientation defects (Fig. 3b, d). Chromosomes were however badly congressed and never aligned in a tight metaphase plate as in control oocytes (Fig. 3c, e). In contrast to a previous study[57], and although chromosomes frequently lagged during anaphase, chromosome segregation was largely accurate in HIM-10-depleted oocytes (Fig. 3f, g). Importantly, we confirmed the lack of significant oocyte aneuploidy following Ndc80 complex loss-of-function through four other approaches: (1) *him-10(RNAi)* alone, (2) *GFP(RNAi)* in a strain carrying endogenously tagged HIM-10::GFP, (3) auxin treatment alone of oocytes expressing endogenously tagged HIM-10::AID, and (4) *ndc-80(RNAi)* in a *him-10* temperature-sensitive strain at the restrictive temperature (26 °C) (Supplementary Fig. 4a–c). Thus, the Ndc80 complex participates in chromosome congression and alignment, but is not essential for accurate chromosome segregation in *C. elegans* oocytes.

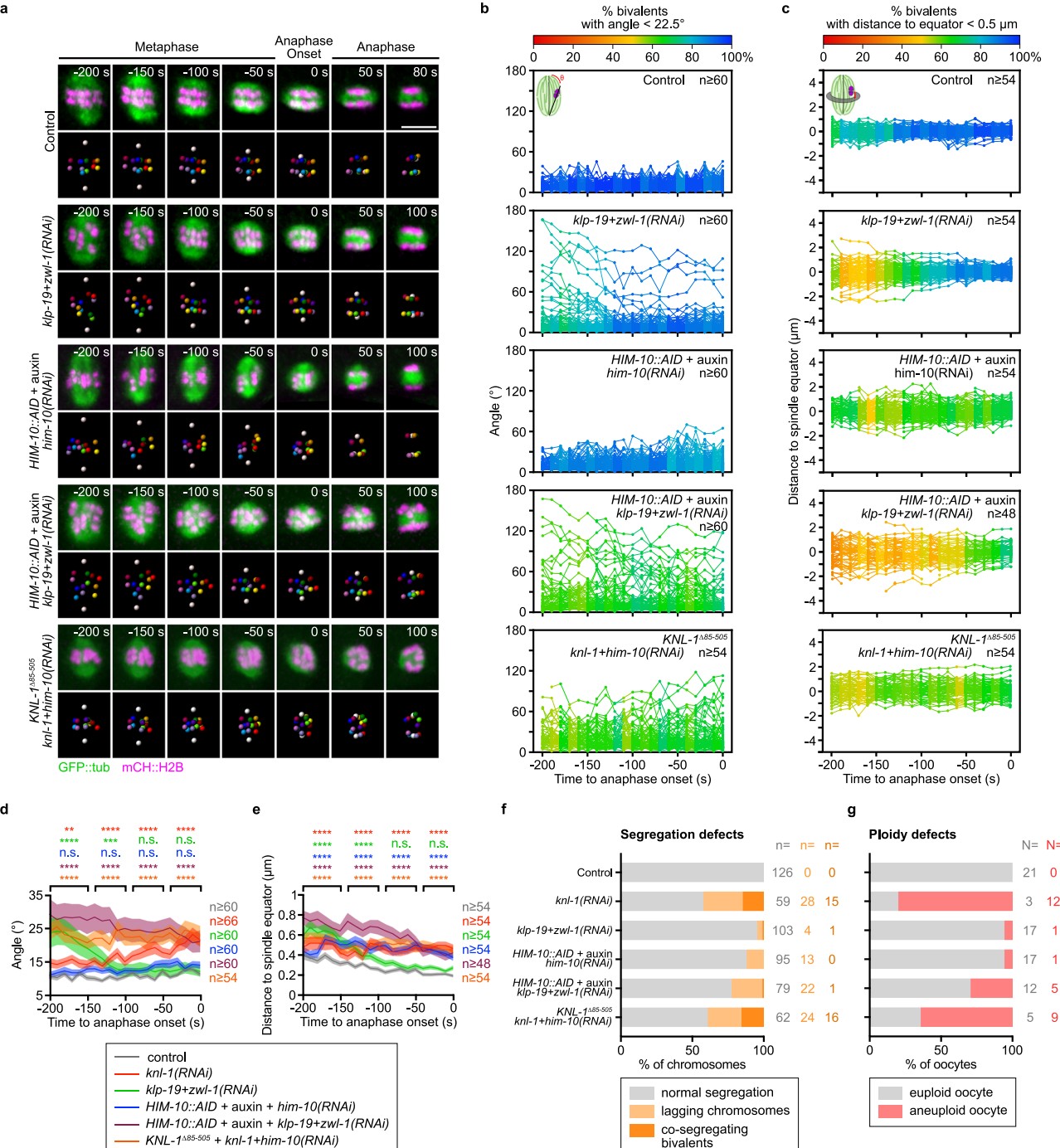

**Fig. 3 | Effects of perturbing kinetochore sub-complexes alone or in combination on chromosome orientation, alignment and segregation.**
**a** Representative time-lapse images, centered on the meiotic spindle and chromosomes, of GFP::TBA-2$^{\alpha\text{-tubulin}}$ or GFP::TBB-2$^{\beta\text{-tubulin}}$ (green) and mCherry::HIS-11$^{H2B}$ (magenta) -expressing oocytes during end of metaphase and anaphase in the indicated conditions. Time relative to anaphase onset is indicated at the top right corner of each panel. Scale bars, 5 μm. **b, c** Plots of individual homologous chromosome pair angles relative to the spindle long axis (**b**) or of the distance between individual homologous chromosome pairs and the spindle equator (**c**) during the 200 s preceding anaphase I onset in oocytes, in the indicated conditions. Each dot corresponds to an individual homologous chromosome pair and is color-coded as indicated at the top of each graph. Schematics of the measured parameter and

sample size (*n* pairs of homologous chromosomes) are at the top left and right corners respectively of each graph. **d, e** Plots of mean angle relative to the spindle long axis (**d**) and distance to the spindle equator (**e**) of the homologous chromosome pairs during the 200 s preceding anaphase I onset in oocytes, in the indicated conditions. Dark lines represent the mean. Lighter bands represent the SEM. Samples sizes (*n* pairs of homologous chromosomes) are at the right of each graph. (one-way ANOVA with Tukey's multiple comparison: n.s., $P \geq 0.05$; **$P < 0.01$; ***$P < 0.001$; and ****$P < 0.0001$). **f, g** Quantification of normally segregating, lagging and co-segregating chromosomes during anaphase (**f**), and euploid or aneuploid oocytes at the end of anaphase (**g**) in indicated conditions. Sample sizes (*n* chromosomes, N oocytes) are at the right of each graph.

## NDC-80 CH domain and the SKA complex establish end-on attachments prior to anaphase onset

Next, we wanted to determine the molecular mechanisms of Ndc80 complex function at kinetochores that control chromosome congression and alignment. A previous analysis showed that chromosomes are stretched in a kinetochore-dependent manner prior to anaphase onset in *C. elegans* oocytes[57]. This stretching can probably be attributed to kinetochore-dependent end-on attachments pulling on holocentric cup-shaped kinetochores to promote chromosome bi-orientation. To determine the exact kinetics of chromosome stretching, and thus of end-on attachment establishments, we monitored homologous chromosome and bivalent length, as well as the inter-homolog distance, over time until anaphase onset. All three chromosomal metrics showed a similar trend with a progressive increase

starting around 150 s and until 50 s prior to anaphase onset (Fig. 4a–c). Consistent with Ndc80-dependent pulling on kinetochores being responsible for this stretching, we found that depleting HIM-10 abrogated the increase in bivalent length observed prior to anaphase onset (Fig. 4d). In contrast with a previous study based on fixed oocytes, this observation is compatible with establishment of Ndc80-dependent end-on connections prior to anaphase I onset in *C. elegans* oocytes[56]. To directly test this assumption, we treated metaphase I-arrested oocytes with a low dose of nocodazole. This treatment highlighted a population of stable microtubules, which greatly decreased in HIM-10-depleted oocytes, but not in absence of dynein-mediated lateral interactions at kinetochores (Fig. 4e, f). Thus, stable end-on attachments are formed prior to anaphase I onset in an Ndc80 complex-dependent fashion in the *C. elegans* oocyte.

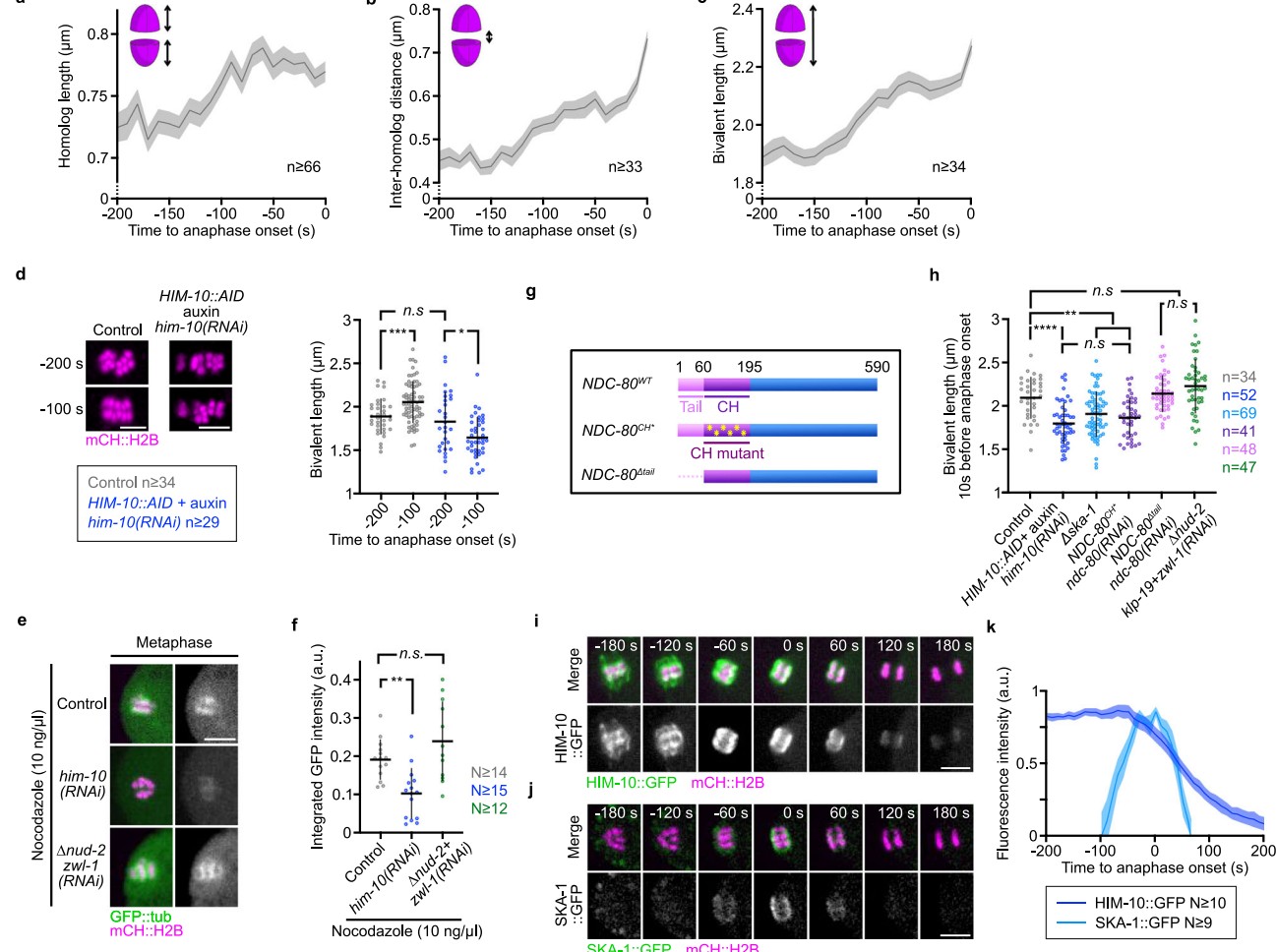

**Fig. 4 | End-on connections apply tension on chromosomes via the NDC-80 CH domain and the SKA complex. a–c** Plots of homologous chromosome length (**a**), distance between homologous chromosomes (**b**) and bivalent chromosome length (**c**) during 200 s preceding anaphase I onset. Schematics of each measurement and sample sizes (n chromosomes) are at the upper left and bottom right respectively of each graph. Dark lines, mean; lighter bands, SEM. **d** Left: Representative images of chromosomes in mCherry::HIS-11[H2B]-expressing (magenta) oocytes in indicated conditions. Scale bar, 5 μm. Right: Plots of bivalent chromosome length 200 s and 100 s before anaphase onset in indicated conditions. Error bars, SD. Sample sizes (n chromosomes) are listed in the key at the bottom left. (t-tests with Welch's correction, two-sided: n.s., $P \geq 0.05$; **$P < 0.01$; ***$P < 0.001$; and ****$P < 0.0001$). **e** Representative images of GFP::TBB-2[β-tubulin] (green) and mCherry::HIS-58[H2B] (magenta) -expressing oocytes treated with 10 ng/μl of nocodazole during metaphase in indicated conditions. Scale bar, 5 μm. **f** Quantification of the integrated GFP::TBB-2[β-tubulin] intensity around chromosomes in oocytes treated with 10 ng/μl of

nocodazole during metaphase in indicated conditions. Error bars, SD. Samples sizes (N oocytes) are at the right of the graph. (one-way ANOVA with Tukey's multiple comparison: n.s., $P \geq 0.05$; **$P < 0.01$; ***$P < 0.001$; and ****$P < 0.0001$). **g** Schematics of wild-type (WT) NDC-80, the CH domain mutant (NDC-80 CH*) and of the tail-deleted mutant (NDC-80 Δtail). (**h**) Plot of bivalent chromosome length 10 s before anaphase onset in indicated conditions. Errors bars, SD. Samples size (n chromosomes) are at the right of the graph. (one-way ANOVA with Tukey's multiple comparison: n.s., $P \geq 0.05$; **$P < 0.01$; ***$P < 0.001$; and ****$P < 0.0001$).
**i, j** Representative time-lapse images of an oocyte expressing endogenously GFP-tagged HIM-10[Nuf2] (**i**, green) or SKA-1[Ska1] (**j**, green) and mCherry::HIS-11[H2B] (magenta) during the first meiotic division. Time relative to anaphase onset. Scale bars, 5 μm.
**k** Quantifications of GFP-tagged HIM-10[Nuf2] and SKA-1[Ska1] intensities on chromosomes over time. Dark lines, mean; lighter bands, SEM. Sample sizes (N oocytes) are in the key.

The microtubule-binding capacity of the Ndc80 complex resides in two distinct elements inside the NDC-80 subunit. A calponin-homology (CH) domain that contacts the microtubule lattice and a basic unstructured N-terminal tail that is proposed to interact electrostatically with the negatively charged tubulin surface[10]. During mitosis in *C. elegans* zygotes, deleting the NDC-80 tail does not perturb chromosome segregation, while mutating the CH domain, to prevent its interaction with microtubules, recapitulates the full NDC-80 loss-of-function phenotype[11,16]. To determine which NDC-80 element is responsible for end-on attachments in *C. elegans* oocytes, we compared bivalent length in control, in HIM-10-depleted oocytes and in oocytes expressing RNAi-resistant transgenes carrying either an NDC-80 CH domain mutant (*NDC-80 CH**), which cannot interact with microtubules, or an NDC-80 tail-deleted mutant (*NDC-80 Δtail*)[11,16] (Fig. 4g). Depleting HIM-10 impaired end-on attachments and led to a significant decrease in bivalent length compared to controls (Fig. 4h). A reduction in bivalent length was also observed upon expression of the NDC-80 CH mutant, but not in the tail-deleted mutant or following inhibition of lateral connection (Fig. 4h). Therefore, as in mitosis, the CH domain of NDC-80 is the primary determinant of end-on kinetochore-microtubule attachments in *C. elegans* oocytes.

In control oocytes, the temporal pattern of chromosome stretching, at least partially, correlated with the progressive loading of the SKA (Spindle and Kinetochore-Associated) complex at kinetochores (Fig. 4i–k). During mitosis, the heterotrimeric SKA complex (comprised of two SKA-1 and one SKA-3 protein in *C. elegans*) functions with the Ndc80 complex to reinforce load-bearing end-on connections between spindle microtubules and kinetochores[14,16]. Consistent with kinetochore-localized SKA complex also participating in the reinforcement of Ndc80-mediated end-on attachments in *C. elegans* oocytes, we measured reduced bivalent lengths in a *ska-1* deletion (*Δska-1*) mutant just prior to anaphase, but not during prometaphase[16] (Fig. 4h and Supplementary Fig. 5a). In contrast to Ndc80 full loss-of-function, chromosome alignment was also only significantly defective during the last 50 s that preceded anaphase onset in the *Δska-1* mutant, which is consistent with the loading kinetics of the SKA complex at kinetochores (Fig. 4i–k and Supplementary Fig. 5b–f). Overall, our results show that the Ndc80 and SKA complexes coordinately promote end-on load bearing attachments at the cup-shaped kinetochores of meiosis I. However, while the Ndc80 complex is present and required throughout meiosis I, the SKA complex contribution to these attachments is limited to the end of prometaphase/metaphase. Nevertheless, these end-on attachments are not essential for accurate chromosome segregation during meiosis I as the vast majority of Ndc80 loss-of-function or *ska-1*-deleted oocytes generated euploid meiosis II oocytes (Supplementary Fig. 5b–h). Therefore, unlike in monocentric species oocytes, chromosome bi-orientation is not a pre-requisite for accurate chromosome segregation during *C. elegans* meiosis[60].

## Unlike in mitosis, end-on connections and lateral interactions are insufficient to account for the essential function of kinetochores

We next tested potential redundancy between lateral and end-on connections for accurate chromosome segregation. Indeed, during mitosis in *C. elegans* zygotes, the kinetochore-null (KNL) phenotype, with unaligned and missegregated chromosomes, observed upon KNL-1 depletion, can be recapitulated by co-depleting downstream Ndc80 and RZZ complexes[70]. To determine if that was also the case in oocytes, we co-depleted HIM-10 and ZWL-1 by RNAi in absence of *nud-2* (Supplementary Fig. 2e). As for the single depletion of HIM-10, simultaneously preventing kinetochore dynein recruitment did not induce any detectable chromosome orientation defects, but chromosomes were badly congressed and never fully aligned (Supplementary Fig. 2c–g). Chromosome segregation was however largely accurate in this condition (Supplementary Fig. 3a, b). We then tested the

additional potential effect of removing all lateral interactions and end-on connections by combining auxin-induced degradation of HIM-10 with simultaneous depletion of ZWL-1 and KLP-19 by RNAi (Fig. 3a, fourth panel). This led to the strongest phenotype of all perturbations tested, with chromosomes displaying a high angle throughout meiosis I and alignment defects comparable to KNL-1-depleted oocytes at anaphase onset (Fig. 3b–e). The lack of chromosome orientation improvement over time in this condition, as compared to inhibition of lateral connections only, suggested that Ndc80-mediated end-on connections are the primary compensatory mechanism. However, unexpectedly and unlike in KNL-1-depleted oocytes, the strong chromosome orientation phenotype was not accompanied by co-segregating bivalents (Fig. 3f). Instead, the vast majority of meiosis I oocytes lacking both lateral and end-on connections gave rise to euploid meiosis II oocytes (Fig. 3g). Thus, we conclude that, unlike in mitosis in zygotes, the functions of the Ndc80 and RZZ complexes, but also of the chromokinesin KLP-19, are insufficient to account for the essential role of KNL-1, and thus of kinetochores, in the control of chromosome segregation accuracy.

## The Ndc80 complex and BHC module act redundantly to promote chromosome proper orientation, congression, and accurate segregation

Next, to specifically assay the kinetochore function of the BHC module, we used an RNAi-resistant deletion transgene of KNL-1 that lacks all MELT repeats (*KNL-1^{Δ85-505}*), and thus is unable to recruit the BHC module at kinetochores upon depletion of endogenous KNL-1[33,71,72] (Supplementary Fig. 6a–d). Oocytes expressing this mutant transgene and depleted of endogenous KNL-1 displayed mild chromosome orientation defects, but overall behaved as controls, with only a single lagging chromosome detected and no associated oocyte aneuploidy (Supplementary Fig. 3a, b and Supplementary Fig. 6e–i). Therefore, although BHC module components are essential for accurate chromosome segregation, their kinetochore localization is not[33,45,46]. To analyze potential functional redundancy between the BHC module at kinetochores and Ndc80 complex, we co-depleted KNL-1 and HIM-10 in *KNL-1^{Δ85-505}*-expressing oocytes (Fig. 3a, fifth panel). Surprisingly, chromosome orientation and alignment were severely perturbed in this condition (Fig. 3b–e). Furthermore, in contrast to the simultaneous inhibition of end-on and lateral connections, frequent co-segregating bivalents, leading to a proportion of aneuploid meiosis II oocytes comparable to KNL-1 depletion, were observed (Fig. 3f, g). We importantly verified that dynein was properly localized at kinetochores in *KNL-1^{Δ85-505}*-expressing oocytes following KNL-1 depletion (Supplementary Fig. 6j–l). Thus, redundant activities of the Ndc80 complex and kinetochore-localized BHC module, recruited downstream of the kinetochore scaffold protein KNL-1, are critical for accurate chromosome orientation and congression during oocyte meiosis in *C. elegans*.

## The BHC module initiates pushing forces at anaphase onset

We next wanted to determine the molecular mechanism of BHC module function at kinetochores. BHC module components control kinetochore microtubule dynamics during metaphase and promote central spindle microtubule assembly in anaphase[33,73]. We thus envisioned two plausible scenarios for the role of the BHC module at kinetochores. First, by controlling microtubule dynamics, BHC components could participate in the generation of efficient end-on attachments, and thus act partially redundantly with the Ndc80 complex. Second, by promoting anaphase central spindle assembly through CLS-2 activity, the BHC module could be essential to generate initial pushing forces at anaphase onset, required for initial chromosome physical separation. To test these hypotheses, we quantified CLS-2::GFP and microtubule density in the mid-bivalent (the region between the two homologous chromosomes of each pair) and central spindle regions over time (Fig. 5a–d). In *KNL-1^{Δ85-505}*-expressing oocytes,

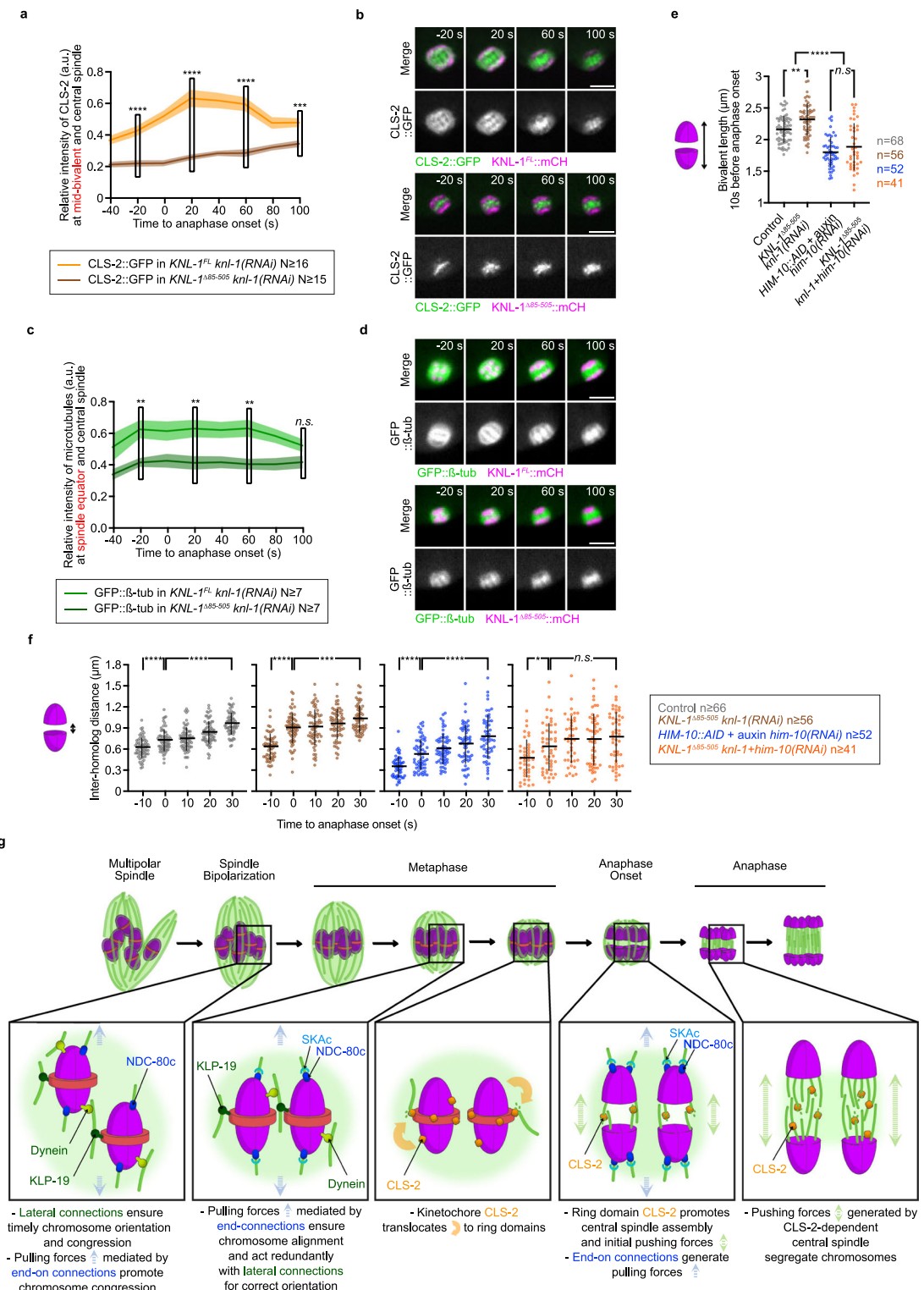

**Fig. 5 | Kinetochore-localized CLS-2, essential to establish initial anaphase pushing forces, cooperates with Ndc80c-mediated pulling to control accurate chromosome segregation. a, c** Quantification of endogenously GFP-tagged CLS-2[Clasp2] (**a**) and GFP::TBB-2[β-tubulin] (**c**) intensities at the mid-bivalent region and central spindle during end of metaphase and anaphase respectively. Dark lines represent the mean. Lighter bands represent the SEM Sample sizes (N oocytes) are indicated below. (t-test with Welch's correction, two-sided: n.s., $P \geq 0.05$; \*\*$P < 0.01$; \*\*\*$P < 0.001$; and \*\*\*\*$P < 0.0001$). **b, d** Representative time-lapse images of an oocyte expressing endogenously GFP-tagged CLS-2[Clasp2] (**b**, green) or TBB-2[β-tubuline] (**d**, green) and KNL-1[Spc105]::mCherry (magenta) during the first meiotic division. Time relative to anaphase onset is indicated at the top right corner of upper panels. Scale bars, 5 μm. **e** Plot of bivalent chromosome length 10 s before anaphase onset in the indicated conditions. Errors bars represent the SD. Samples size (*n* pairs of homologous chromosomes) are at the right of the graph. (one-way ANOVA with Tukey's multiple comparison: n.s., $P \geq 0.05$; \*\*$P < 0.01$; \*\*\*$P < 0.001$; and \*\*\*\*$P < 0.0001$). **f** Plots of the distance between homologous chromosomes between 10 s before and 30 s after anaphase onset in the indicated conditions. Errors bars represent the SD. Sample sizes (*n* chromosomes) are in the key. (t-test with Welch's correction, two-sided: n.s., $P \geq 0.05$; \*\*$P < 0.01$; \*\*\*$P < 0.001$; and \*\*\*\*$P < 0.0001$). **g** A model for holocentric kinetochore sub-complexes function in chromosome orientation, alignment and accurate chromosome segregation.

following depletion of endogenous KNL-1, CLS-2::GFP intensity was decreased at the mid-bivalent and central spindle region compared to controls throughout meiosis I (even before anaphase onset). These reduced CLS-2::GFP levels correlated with decreased microtubule density throughout meiosis I (Fig. 5a, c). The decreased microtubule density prior to anaphase in absence of BHC at kinetochores was compatible with the first hypothesis. However, if anything, bivalent stretching seemed slightly but significantly increased in this condition, and only decreased upon simultaneous depletion of HIM-10 (Fig. 5e). This is inconsistent with a role for BHC components in participating in the generation of efficient pulling forces via end-on attachments. Quantification of CLS-2::GFP and microtubule density over time showed that, as in mitosis, kinetochore targeting of CLS-2 was also essential for its efficient accumulation at central spindle and for full central spindle microtubule assembly in anaphase[73] (Fig. 5a–d). To determine if the reduced central spindle microtubule density during early anaphase could affect initial chromosome physical separation, we measured the distance between each homolog of a pair (inter-homolog distance) over time in absence of kinetochore targeting of BHC components, or following HIM-10 depletion, or both (Fig. 5f). As expected, in control oocytes, the inter-homolog distance increased progressively over time after anaphase onset. Similar increase of the inter-homolog distance was evident in HIM-10-depleted oocytes, albeit with a lower starting inter-homolog distance due to reduced pulling forces in this condition. In absence of BHC kinetochore targeting, the inter-homolog distance also increased during the first 30 s following anaphase onset, probably through Ndc80-mediated pulling forces, but with a slower pace compared to control or HIM-10-depleted oocytes. In contrast, following an initial jump in the inter-homolog distance due to loss of homologous chromosome cohesion at anaphase onset, the inter-homolog distance stayed relatively constant in absence of both BHC module kinetochore targeting and HIM-10. Thus, our results support a model where kinetochore-localized BHC module components control initial chromosome physical separation at anaphase onset by promoting efficient central spindle microtubule assembly, and thus promote initial pushing forces on chromosomes (Fig. 5g). Overall, our results are consistent with initial pushing forces being important for accurate chromosome segregation, as their inhibition leads to severe oocyte aneuploidy when Ndc80-mediated pulling forces are also absent.

## Discussion

We had previously shown that, in *C. elegans* oocytes, kinetochores are essential for chromosome segregation accuracy, but not for their physical separation[45]. Instead, we demonstrated that an atypical central spindle-dependent pushing mechanism drives chromosome movements during anaphase in this system[46]. Here we further show that initial pushing forces are also important, redundantly with Ndc80 complex-mediated pulling, for the control of chromosome segregation accuracy. The present study not only extends our understanding of the kinetochore-dependent control of segregation accuracy, but also reconciles apparently conflicting conclusions (discussed below) from previous work in *C. elegans* oocytes.

Kinetochores provide two main modes of interaction with spindle microtubules. Kinetochore dynein promotes lateral interactions with the microtubule lattice, while the Ndc80 and SKA complexes cooperatively promote end-on load bearing attachments to microtubule plus-ends[6]. In parallel, the microtubule motor chromokinesin KLP-19 generates polar ejection forces that push chromosomes toward the spindle equator[61,65]. Previous conflicting studies in *C. elegans* oocytes had concluded on the exclusive requirement for lateral or end-on microtubule connections in the control of chromosome alignment and segregation[46,55–57,62,74]. Our results reconcile these seemingly contradictory results. Indeed, we show that both types of microtubule-based interactions participate in timely and/or proper chromosome

orientation and congression. Our results are consistent with a model in which initial lateral interactions between kinetochores and microtubules, provided by the redundant activities of the chromokinesin KLP-19 and dynein, promote timely chromosome orientation and congression along the spindle long axis. The fact that plus and minus-end directed motors could act in a redundant manner might sound counterintuitive at first. However, the peculiar microtubule organization within the meiotic spindle of *C. elegans* oocytes, which is comprised of numerous short microtubules with no preferred polarity relative to the spindle long axis, likely underlies the redundancy of opposite polarity motors[75]. While previous work concluded on the essential function of lateral connections mediated by KLP-19 and/or dynein, we show here that meiotic chromosome segregation is largely accurate in their absence in *C. elegans* oocytes[55,62,63]. We attribute these inconsistent conclusions to the different approaches used for analyzing the contributions of microtubule motors in chromosome movements. Indeed, we analyzed chromosome segregation in intact bipolar spindles, while previous studies focused on chromosome movements on experimentally-induced monopolar spindles, which may not accurately reflect wild-type chromosome-microtubule interactions.

During mitosis in *C. elegans* zygotes, the kinetochore-null phenotype obtained upon depletion of the centromeric histone H3 variant CENP-A, or of kinetochore scaffold components such as CENP-C, KNL-1 or 3, can be recapitulated by simultaneously depleting downstream kinetochore components of the Ndc80 and RZZ complexes[70]. This highlights the simultaneous requirement for Ndc80 complex-mediated end-on load bearing attachments, and dynein-dependent lateral interactions between kinetochores and spindle microtubules, during mitosis. In contrast during meiosis, our present results place the BHC module and Ndc80 complex as major, and redundant, contributors of chromosome segregation accuracy in *C. elegans* oocytes. This is in agreement with a recent study that reported Ndc80 complex-dependent stretching of meiotic chromosomes prior to anaphase onset in *C. elegans* oocytes[57]. Our results further demonstrate that this chromosomal stretching also depends on the SKA complex toward the end of prometaphase and during metaphase, and is the consequence of end-on attachments to microtubules. However, in contrast to the earlier study[57], which attributed a central role to the Ndc80 complex alone in the physical separation of chromosomes, our results show that neither the Ndc80 complex, nor BHC module kinetochore targeting, are necessary on their own for accurate chromosome segregation or physical separation. This highlights the functional redundancy of the Ndc80 complex and of the BHC module at kinetochores, in the control of chromosome segregation accuracy in *C. elegans* oocytes. The discrepancy about Ndc80 complex requirement for chromosome physical separation between the previous and our present study is again likely due to the different approaches used. Indeed, we found that combining *in utero* microscopy of oocytes (method used in the previous study) with protein perturbations severely aggravated chromosome segregation phenotypes as compared to the same perturbations filmed *ex utero* (method we used in this study, Fig. 2c, f, g and Supplementary Fig. 1c–e). We attribute this phototoxicity to two factors specifically associated with *in utero* live imaging. First, filming *in utero* exposes oocytes to an overall higher load of energy due to longer exposure of oocytes to laser excitation (filming starts at or even before NEBD) and to higher light excitation required to cross the worm cuticle and overlying tissues. As experimental perturbations, such as RNAi treatments, can increase the specimen's sensitivity to light exposure, aggravated phenotypes can be caused by phototoxic effects rather than specific ones[76]. Second, the experimental montage required for *in utero* live imaging places worms, and oocytes, in hypoxic conditions. Previous work in *C. elegans* embryos demonstrated that even short exposure to oxygen deprivation can cause aberrant chromosome segregation[77]. Overall, we conclude that the aggravated effects of Ndc80 complex depletion in the

previous study are likely due to the increased toxicity of *in utero* microscopy.

We must note that in absence of both the Ndc80 complex and BHC module at kinetochores, chromosome orientation and alignment are perturbed throughout meiosis I although KLP-19 and dynein are present on chromosomes (Fig. 3a–e). This result is in apparent contradiction with the role we attribute to these two microtubule motors in redundantly promoting timely chromosome orientation and congression. We suspect that, although productive end-on attachments can be established in absence of kinetochore-localized BHC module (Fig. 5e), lateral interactions might not function as efficiently in this condition. The combination of low-efficiency lateral interactions in absence of kinetochore-localized BHC module, combined with the lack of end-on attachments upon HIM-10 depletion, could account for the defective chromosome orientation and alignment observed in this condition.

We previously showed that the BHC module redistributes from the kinetochore to the anaphase central spindle during oocyte meiosis and mitosis in *C. elegans*[46,73,78]. This in turn promotes central spindle microtubule assembly through the activity of the microtubule-stabilizing protein CLS-2. During oocyte meiosis, the CLS-2-dependent central spindle assembly is the primary driver of chromosome physical separation during anaphase[46,54]. Here we extend the role of kinetochore-localized BHC components (and thus of CLS-2) to the control of chromosome segregation accuracy. We provide evidence that kinetochore-localized CLS-2 is required for initial pushing of homologous chromosomes during early anaphase. We suspect that kinetochore-localized CLS-2 translocates from the kinetochores to concentrate at the mid-bivalent region, and that this relocalization initiates before anaphase onset as evidenced by quantifying GFP-tagged CLS-2 on chromosomes over time (Supplementary Fig. 6b, m). We propose a model where the initial CLS-2-dependent pushing acts redundantly with Ndc80 complex-mediated pulling for accurate chromosome segregation. Previous work concluded on the presence of inter-chromosomal microtubule-free channels during early anaphase, which is inconsistent with the model we propose[55,56]. The existence of these channels was based on observations of light microscopic data or of electron tomographic reconstructions of fixed cells. Although regions of low microtubule density are visible between homologous chromosomes during early anaphase in both types of observations, they do not provide evidence to exclude that microtubules could initially push on the periphery of the mid-bivalent region of homologous chromosomes. We therefore envision a mechanism where both pulling and pushing can redundantly provide the initial separation of homologous chromosomes, necessary to avoid their eventual co-segregation to the same daughter cell. In absence of both mechanisms, homologous chromosomes remain closely apposed after anaphase onset and have a higher tendency to co-segregate, especially when initially mis-oriented relative to the spindle long axis.

Consistent with this view, we found a strong correlation between the angle of chromosomes at anaphase onset and their behavior during anaphase (Supplementary Fig. 7a, b). Our results show that while the chromosome angle is not an absolute predictor of the outcome of segregation, a strong correlation exist between the probability of a chromosome ending up mis-segregated in anaphase and higher angles relative to the spindle long axis in metaphase. By extension, we conclude that a prime function of meiotic kinetochores in *C. elegans* oocytes is to orient chromosomes relative to the spindle axis in metaphase in order to avoid segregation errors in anaphase. Whether this function is a common feature of holocentric species, or could even be broadly shared with monocentric species is unknown as the role of individual kinetochore sub-complexes has not been thoroughly investigated in oocytes of other holocentric or monocentric species. This should be the topic of future studies. Together, our results provide a unified view of the mechanisms that control holocentric chromosome segregation accuracy and their physical separation in *C. elegans* oocytes.

## Methods

### *C. elegans* strain maintenance
The strains used in this study are listed in Supplementary Table 1. Strains were maintained on nematode growth medium (51 mM NaCl, 2,5 g Bacto Peptone, 17 g Bacto Agar, 12 μM cholesterol, 1 mM CaCl₂, 1 mM MgSO₄, 25 μM KH₂PO₄ and 5 μM Nystatin in a 1 L ddH2O final) plates seeded with OP50 bacteria at 23 °C, except JDU537 (*him-10(e511ts)*), which was maintained at 16 °C. All strains were generated by crossing pre-existing strains or by CRISPR/Cas9-mediated insertion.

### RNA-mediated interference
PCR reactions were cleaned (QIAquick Gel Extraction Kit, Qiagen #28704) and used as templates for T3 and T7 transcription reactions (MEGAscript transcription kit, Invitrogen #AM1334 for T7 and #AM1338 for T3), which were cleaned (MEGAclear transcription cleanup kit, Invitrogen #AM1908) and eluted in 50 μL of RNase-free water. Single stranded RNAs were combined and annealed at 68 °C for 10 min, followed by 37 °C for 30 min. Templates and primers used to synthesize dsRNA are listed in Supplementary Table 2. L4 hermaphrodites were injected with dsRNAs at the indicated concentrations and incubated at 20 °C, or 16 °C for JDU537 (*him-10(e511ts)*), for 48 h before further processing.

### Auxin-induced degradation
We crossed the strain PHX3493, carrying *him-10* endogenously tagged with an auxin-inducible degron (mAID), with the strain JDW10, expressing a TIR1 transgene (*sun-1p::TIR1::F2A::BFP::AID*::NLS::tbb-2 3'UTR*) under the control of the germline specific sun-1 promoter[79], and then with the GFP::TBB-2$^{\beta-tubulin}$- and mCherry::HIS-11$^{H2B}$, under the control of the germline specific mex-5 promoter, -expressing worms from the strain JDU19. HIM-10 depletion was achieved by plating adult worms from the resulting strain (JDU659), on OP50 seeded plates containing NGM + 1 mM auxin for 1 to 4 h before performing live imaging. HIM-10 auxin-induced depletion was combined with KLP-19 and ZWL-1 co-depletion by RNAi to abrogate all end-on and lateral connections to microtubules, or was combined with *(him-10)RNAi* to ensure complete inhibition of end-on connections.

### Live *in utero* imaging
GFP::TBA-2$^{\alpha-tubulin}$- and mCherry::HIS-11$^{H2B}$-expressing adult worms were anesthetized in M9 (22 mM KH₂PO₄, 42 mM Na₂HPO₄, 86 mM NaCl, 1 mM MgSO₄) with 50 ng/μl of tricaine (Sigma #E10521-10G) and 200 ng/μl of tetramisole (Sigma #T1512-10G) for 30 min[80,81]. Worms were then transferred with an eyelash in 1 μl of M9 on an agarose pad (2% agarose in M9) placed on a glass slide, and an 18 × 18 mm coverslip was placed on top of this montage. The room temperature was monitored during filming and varied between 21 °C-24 °C. All acquisitions were performed using a Nikon Ti-E inverted microscope, equipped with a CSU-X1 (Yokogawa) spinning-disk confocal head with an emission filter wheel and a dual camera system (2x coolSNAP HQ2 CCD camera, Photometrics Scientific). The power of lasers were measured before each experiment with an Ophir VEGA Laser and energy meter. Fine stage control and focus correction during acquisition was performed using a PZ-2000 XYZ Piezo-driven motor from Applied Scientific Instrumentation (ASI). 30 Z-plans, separated by 0.3 μm, were acquired every 10 s using a Nikon CFI APO λS 40x/NA1.25 water immersion objective. Movies were acquired with a 1 × 1 binning. Control of acquisition parameters was done using the Metamorph 7 software (Molecular Devices). Image analysis was performed using the Fiji[82] and Imaris (Oxford Instruments) software.

### Live *ex utero* imaging
GFP::TBA-2$^{\alpha-tubulin}$-, or GFP::TBB-2$^{\beta-tubulin}$-, and mCherry::HIS-11$^{H2B}$-expressing adults worms were dissected in meiosis medium (5 mg/mL inulin, 25 mM HEPES, 60% Leibovitz's L-15 media and 20% fetal bovine)

to free fertilized oocytes. Imaging at 23 °C, or temperature shifts from 16 °C to 26 °C for the *him-10(e511ts)* mutant, were performed using the CherryTemp temperature controller system (CherryBiotech). All films were acquired with the same spinning disk confocal microscope as described previously using a Nikon CFI APO λS 60x/NA1.4 oil immersion objective. Movies were acquired with 2 × 2 binning. For all movies, except Fig. 4e, j, Fig. 5b, d, Supplementary Fig. 2a, b, Supplementary Fig. 4a *him-10(e511ts)* mutant and Supplementary Fig. 6b, c, j, k, 30 Z-plans, separated by 0.3 μm, were acquired every 10 s with the dual camera system. For Fig. 4e, j, Fig. 5d, Supplementary Fig. 2a, b, and Supplementary Fig. 4a *him-10(e511ts)* mutant, 4 Z-plans, separated by 2 μm, were acquired every 20 s using a single camera. For Fig. 5b and Supplementary Fig. 6b, c, 2 Z-plans, separated by 1 μm, were acquired every 20 s using the dual camera system. For Supplementary Fig. 6j, k, 4 Z-plans, separated by 2 μm, were acquired every 10 s using a single camera. Control of acquisition parameters was ensured by the Metamorph 7 software (Molecular Devices). Image analysis was performed using the Fiji[82] and Imaris (Oxford Instruments) software.

### Nocodazole treatment
GFP::TBB-2[β-tubulin]- and mCherry::HIS-58 [H2B]-expressing adult worms were dissected in meiosis medium with 10 ng/μl of nocodazole to depolymerize microtubules in oocytes, which are devoid of egg-shell and are thus permeable. Movies of oocytes in early meiosis were acquired by live *ex utero* imaging as described previously.

### Chromosome and spindle pole spotting and tracking
To improve the signal-to-noise ratio, raw microscopy images were pretreated with mean and median filters in Fiji. Filtered stacks were 3D-projected in Imaris. Chromosomes (mCherry::HIS-11[H2B] signal) were semi-automatically spotted during metaphase and beginning of anaphase using the spot function in Imaris. A threshold diameter of 0.8 μm was used to detect chromosomes. This initial spotting was then manually improved. The spindle poles (GFP::TBA-2[α-tubulin] or GFP::TBB-2[β-tubulin] signal) were manually spotted as the center of each extremity of the spindle long axis. Chromosomes and spindle poles were tracked over time, and their XYZ coordinates were extracted to perform quantitative analyzes.

### Quantitative analyzes
Chromosome orientation was quantified by measuring the angle between each bivalent and the spindle long axis. Average maximum angles of chromosomes in *in utero* control oocytes is 22.5° and was used as a threshold angle for correct chromosome orientation. Chromosome alignment was quantified by measuring the distance of the center of each bivalent from the spindle equator. Average maximum distances from the spindle equator of chromosomes in *in utero* control oocytes is 0.5 μm and was used as a threshold distance for correct chromosome alignment. Chromosome compaction was quantified by measuring the distance of the center of each bivalent from the spindle long axis. Average maximum distances from the spindle long axis of chromosomes in *in utero* control oocytes is 1.6 μm and was used as a threshold for correct chromosome compaction. Chromosome oscillation was quantified by subtracting the distance of the center of each bivalent from the spindle equator to the same distance 10 s later, absolute values are plotted.

### Fluorescence intensity measurements
Fluorescently-tagged protein quantifications for Fig. 4e, f, j, k (GFP::TBB-2[β-tubulin]; SKA-1[Ska1]::GFP), Fig. 5c, d (GFP::TBB-2[β-tubulin]) and Supplementary Fig. 2a, b (DHC-1[Dync1h1]::mNG; mCherry::HIS-11[H2B]) were generated by imaging embryos at 20 s intervals, with 4 Z-plans, separated by 2 μm. For Fig. 4i, k (HIM-10[Nuf2]::GFP), quantifications were generated by imaging embryos at 10 s intervals, with 30 Z-plans, separated by 0.3 μm. For Fig. 5a, b (CLS-2[Clasp2]::GFP) and Supplementary Fig. 6b–d, m (CLS-2[Clasp2]::GFP; KNL-1[Spc105]::mCherry), quantifications

were generated by imaging embryos at 20 s intervals, 2 Z-plans, separated by 1 μm. For Supplementary Fig. 6j–l (DHC-1[Dync1h1]::mNG; KNL-1[Spc105]::mCherry), quantifications were generated by imaging embryos at 10 s intervals, with 4 Z-plans, separated by 2 μm. Measurements were carried out on sum Z-projections for Fig. 4f (GFP::TBB-2[β-tubulin]), Fig. 4k (HIM-10[Nuf2]::GFP; SKA-1[Ska1]::GFP), and Fig. 5c (GFP::TBB-2[β-tubulin]) inside a region of interest including all chromosomes, except for Fig. 5c where a 15-pixel wide linescan covering the entire chromosome length was used. Measurements were performed on a single focal plane for Fig. 5a (CLS-2[Clasp2]::GFP), Supplementary Fig. 2b (DHC-1[Dync1h1]::mNG; mCherry::HIS-11[H2B]) and Supplementary 6d, l, m (CLS-2[Clasp2]::GFP; KNL-1[Spc105]::mCherry; DHC-1 [Dync1h1]::mNG) with one-pixel linescans. Mean background intensities measured in the cytoplasm of oocytes was subtracted to pixel intensities, or mean pixel intensities for Fig. 5c (GFP::TBB-2[β-tubulin]). Normalization was performed by dividing this value by the mean pixel background intensities. Then, for all figures, background-normalized intensities were divided by the maximum pixel intensity of control.

### Chromosome stretching measurements
Chromosome metrics (bivalent length, inter-homolog distance, homolog length) quantifications were performed manually using the measurement point function in Imaris. Measurements were made on 3D-projections at the indicated times.

### Graphs and Statistics
Images and datasets presented in figures are listed in Supplementary Table 3. GraphPad Prism 8 was used to generate all graphs and statistics (mentioned in the figure legends), except for color-coded graphs of chromosome orientation, alignment and axial compaction, which were generated by a custom-made python script. This script is used to plot chromosome orientation, alignment and compaction values of each chromosome every 10 s, and to color-code the proportion of chromosome angles, distances from the spindle equator and distances from the spindle long axis below the thresholds defined previously, at each time points. P-values of all statistical tests performed in this study are listed in Supplementary Table 4.

### Reporting summary
Further information on research design is available in the Nature Portfolio Reporting Summary linked to this article.

## Data availability
All data supporting the findings of this study are available within the paper and its Supplementary Information. Source data for each figure is provided as a separate 'Source Data' Excel file. Source data are provided with this paper.

## Code availability
The Python script developed to generate color-coded graphs of chromosome orientation, alignment and axial compaction was deposited in Zenodo [https://zenodo.org/record/7998311][83].

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

## Acknowledgements

We thank all members of the Dumont lab for support and advice. We are grateful to Patricia Moussounda, Téo Bitaille and Vincent Maupu-Massamba for providing technical support. We thank Vincent Con-tremoulin from the ImagoSeine facility for his technical help with the Imaris Software. We are grateful to Meredith Price for her support with the Imaris Software during the Covid-19 lockdown. We thank Arshad Desai and Dhanya Cheerambathur for providing some *C. elegans* strains used in this study. Some strains were provided by the CGC, which is funded by NIH Office of Research Infrastructure Programs (P40 OD010440). This work was supported by CNRS and University Paris Cité, by NIH R01GM117407 and R01GM130764 (J.C.C.), and by grants from the European Research Council ERC-CoG ChromoSOMe 819179 and from the Agence Nationale de la Recherche ANR-19-CE13-0015 (J.D.).

## Author contributions

M.A., L.P.-N. and J.D. conceived of the project. K.L. performed pilot experiments on 3D filming and tracking in C. elegans oocytes. M.A. and L.P.-N. performed all other experiments and data analysis. A.P. devel-oped the Python script used to generate color-coded graphs. T.G. and H.G. provided technical support. M.A., L.P.-N. and J.D. designed all

experiments. J.C.C. and J.D. made intellectual contributions. M.A. and J.D.t made the figures and wrote the manuscript.

## Competing interests

The authors declare no competing interests.
