## [Peer review file · Nature Communications]

REVIEWER COMMENTS

Reviewer #1 (Remarks to the Author):

In the present manuscript, Pitayu-Nugroho, Aubry et al from Julien Dumont's group address the role of different kinetochore components and microtubule associated proteins during oocyte meiosis in the holocentric nematode *C. elegans*.

There has been a gap in our knowledge around the role of kinetochore components during *C. elegans* oocyte meiosis and the current work establishes a new analysis pipeline that allows the tracking of individual homologous chromosomes during Meiosis I. This level of resolution provides new information that was not attainable in previous analyses. Above all, it provides unequivocal insight into the ploidy state of the oocyte, which is key because previous studies focused on the spatial separation of two chromosome masses without a proper understanding of what that DNA content was. Therefore, the analysis can identify aneuploid oocytes that had seemingly segregated their homologs without problems. Taking advantage of this, the authors show that homologous chromosomes displaying higher orientation defects are more likely to co-segregate.

Pitayu-Nugroho, Aubry et al set to dissect the contribution of different kinetochore/chromosome proteins/complexes in chromosome orientation, congression, and correct segregation. To this end, the authors focus on 1) HIM-10, part of the NDC-80 complex; 2) SKA-1, part of the NDC-80-associated SKA complex; 3) ZWL-1, part of the RZZ complex, which recruits the motor Dynein to kinetochores; 4) the chromokinesin KLP-19; and 5) the CLASP orthologue CLS-2. They conclude that holocentric chromosome bi-orientation is not strictly required for accurate chromosome segregation. Instead, they suggest that initial kinetochore-localized CLS-2-dependent pushing acts redundantly with Ndc80 complex-mediated pulling for accurate chromosome segregation in meiosis.

In summary, the current work provides a significant step further in our understanding of meiotic chromosome segregation in the holocentric *C. elegans*. The experiments are well performed, controlled, and the quality of the data is excellent and presented in a very clear way.

I would be very much in favour of publication after some concerns/comments (listed below) are presented to the authors. I will not need to see the manuscript again as the points I raise should be addressed at the authors' discretion.

1. I am not clear what the authors mean by BHC 'module'. They have shown previously that BUB-1 targets HCP-1, which in turn recruits CLS-2. However, we don't know if that is how the complex is composed *in vivo*. Importantly, these proteins don't share 100% localisation, so I felt it was at times misleading to refer to BHC to functions attributed to either BUB-1, HCP-1, or CLS-2. In this manuscript for example, the authors show regulation of CLS-2 localisation and still refer to the results as BHC module.

2. I got slightly confused as to when the authors use *in utero* and *ex utero* imaging. Could that be further clarified?

3. While the authors mention that they 'analysed the contribution of the main chromosomal components involved in contacts with microtubules', key proteins like KLP-7 (MCAK) which are present in ring domains and in chromosome/kinetochore are not mentioned or analysed. This is somewhat surprising given previous work from the group showing that KLP-7 is key during meiosis I.

4. It appears that KLP-19 and ZWL-1 play a role in orientation at early stages (i.e. Prometaphase I, Figure 3a,b,d), while KNL-1 depletion affects later stages, closer to Metaphase I (Figure 2d). What seems to be an addition of the two effects is achieved in two conditions, as summarised in Figure 3d: 1) KLP-19 and ZWL-1 + HIM-10 depletion (orange line) and 2) *knl-1^{Δ85-505}* +

HIM-10 depletion.

a. In KLP-19/ZWL-1/HIM-10 depletion, I presume KNL-1 is still there and likely BUB-1 and CLS-2 as well, is this correct?

b. For *knl-1^{Δ85-505}*/HIM-10 depletion, I believe KLP-19 should still be present in the ring domain, along with BUB-1 and CLS-2 (Macaisne et al). Do the authors think that KLP-19 is not able to sustain lateral attachments in this condition? Does RZZ/Dynein localise normally in the *knl-1^{Δ85-505}* mutant?

I might be missing something here, but it would be worth explaining these results and what the authors interpret from them in a more detailed way.

5. The authors use RZZ component depletion to remove kinetochore Dynein (which is clearly shown). Does non-kinetochore Dynein, which seems present in the spindle area and next to chromosomes from Metaphase, play a role in orientation and/or congression? Have the authors analysed the effect of KLP-19 and Dynein co-depletion (not KLP-19 and RZZ)?

6. Figures 4a-c could be slightly and inadvertently misleading by the fact that they do not start at zero. I think the zooming is important to show the trend, but it also gives the false impression of 'proximity to zero'.

7. How was the 'low' dose of nocodazole established?

8. In Figure 4g, do the different transgenes express to similar levels?

9. While the discussion and model explain that SKA participates in end-on attachments (bivalent stretching) closer to Metaphase I, this is not so clear in the text, where a focus on the timescale could be emphasised. Also, if the analysis shown in 4h was done at 100 s before anaphase onset (instead of 10 s), does the $\Delta ska1$ still have an effect? If so, it would be difficult to attribute to its kinetochore localisation.

10. In the subheading and text related to Figure 5, it would be more accurate to refer to CLS-2 rather than the 'BHC module', since BUB-1 and HCP-1 are not analysed, and they could be playing other roles as well.

11. Also related to CLS-2, it appears that *knl-1^{Δ85-505}* leads to a re-localisation of CLS-2, rather than or in addition to a reduction in intensity. In fact, the midbivalent CLS-2 localisation is clearer in the *knl-1^{Δ85-505}* mutant. Hence, the statements in lines 391-393 seems like an over-simplification of what's happening.

Federico Pelisch

Reviewer #2 (Remarks to the Author):

Correct chromosome segregation in mitosis and meiosis is the basis for continued cell proliferation and gamete production, respectively. During cell division, accurate segregation of the chromosomes depends on the interaction between spindle microtubules and kinetochores to ensure timely chromosome congression and orientation, and alignment at the metaphase plate. While the molecular players at kinetochores and their function during mitotic cell divisions are well understood, our knowledge of how faithful chromosome segregation during meiosis is achieved is incomplete. As

accurate chromosome segregation in female gametes is critical to avoid the generation of aneuploid eggs, progress in understanding how this is achieved, is seminal. Because of the difficulties associated with studying meiosis in mammalian oocytes, model systems such as *D. melanogaster* and *C. elegans*, as used here, are important research tools for the analysis of meiosis.

In this manuscript, Pitayyu-Nugroho and colleagues use a high-resolution 4D live imaging methodology to track the six bivalent chromosomes and meiotic spindle of *C. elegans* oocytes to analyse chromosome segregation in *C. elegans* oocytes. Through the systematic depletion of kinetochore proteins, the authors reveal the roles of each kinetochore component, exposing critical differences in the mechanisms of accurate chromosome segregation between organisms with holocentric and monocentric chromosomes. They show that initial lateral interactions between kinetochores and microtubules promote timely chromosome orientation and congression, while end-on attachments were observed to be particularly critical for chromosome alignment but not separation. Interestingly, in holocentric *C. elegans* oocytes there does not seem to be a requirement for bi-orientation before chromosome segregation to achieve correct segregation. At the molecular level, the NDC80 complex, the RZZ complex and the *C. elegans* KIF4A homologue klp-19 all contribute to meiotic chromosome segregation, but even when disturbed in combination do not give the same severe chromosome segregation phenotype as depletion of KNL1. However, combining the loss of NDC80 and the KNL1-associated BHC module, containing BUB1, as well as CENP-F and CLASP homologues, results in severe perturbation of chromosome segregation, similar to loss of KNL1. This leads the authors to conclude that pushing forces, generated by the BHC complex, and pulling forces, created by the NDC80 complex, are required together to achieve accurate chromosome segregation in *C. elegans* oocytes. The manuscript advances our understanding of meiotic chromosome segregation, and on the whole, the study is carefully conducted and the results are convincing. There are some issues, though, that should be addressed before publication.

Major points:

- The authors suggest a dominant role for the BHC complex in generating pushing forces required for correct chromosome segregation. However, at no point in the study, is the BHC complex directly analysed; in all experiments where the goal is to study the function of the BHC complex, the version of KNL1 which lacks all MELT motifs is employed. It should be demonstrated that the observed defect is indeed due to the loss of the BHC complex and not some other defect connected to the loss of this region in KNL1.
- When KNL-1 is replaced by KNL-1 Δ 85-505 without MELT motifs, is it clear that BUB1, HCP-1/2 and CLS-2 are not at kinetochores anymore? This needs to be demonstrated.
- The interpretation of the data obtained upon the depletion of different kinetochore complexes depends critically on the successful depletion of the proteins of interest. It therefore needs to be demonstrated that all the RNAis are working successfully to the same extent, and differences in phenotype severity are not just due to differences in depletion efficiency. This is particularly pertinent since the authors point out that for the *C. elegans* NDC80 complex varying degrees of depletion obtained by different research groups may have resulted in differential phenotypes observed by these groups.
- Can the authors comment on or provide evidence against the independency of kinetochore proteins? For example, does depleting components of Ndc80 complex affect KNL1 localisation as it does mammalian cells?

Minor points:

- The order of the figures compared to the flow of text is slightly confusing due to the regular return to additional panels of figure 3. Some figure rearrangement may be required.
- Can the authors comment on the minor segregation defects observed for the double KLP-19 and ZWL1 depletion? Data suggests that chromosomes reach a point comparable to controls but a mild increase in mis-segregation is observed.
- Could representative live cell imaging stills be shown for Figure 5a and b, so the reader can evaluate the differences in mitotic progression?

Reviewer #3 (Remarks to the Author):

The manuscript by Pitayu-Nugroho and coworkers provides a systematic analysis of chromosome segregation in *C. elegans* oocytes.

Firstly, they develop a live-imaging assay to image chromosome segregation from its beginning to the end in oocytes. For this, they acquire fast 3D stacks of samples with chromatin (H2B-mCherry) and microtubules (tubulin-EGFP) labeled. They then analyze data using a semi-automated workflow. These analyses provide a highly quantitative characterization of the entire process with impressive resolution. The authors can precisely measure, in a large number of oocytes, chromosome orientation, chromosome alignment, axial compaction. In addition, measuring the angles of bivalents turned out to be an exceptionally informative parameter.

This assay is then used throughout the manuscript and combined with systematic genetic perturbations of kinetochore components. The authors show that KLP-19 and dynein-/dynactin-mediated lateral interactions promoted timely chromosome orientation and congression in prometaphase. They further show the requirement for the Ndc80 complex for generating pulling forces through end-on attachments, mediating chromosome alignment in metaphase. Interestingly, neither of these two complexes is strictly required for accurate chromosome segregation. Chromosome segregation is instead primarily mediated by the BHC module (composed of BUB-1, CENP-F and CLASP), which facilitates central spindle assembly and generates pushing forces to separate kinetochores in anaphase.

The manuscript is beautifully illustrated with figures that visualize imaging data as well as quantitative analyses in a very intuitive way. This is further helped by excellent schemes, which make it easy to follow even complex perturbations and their consequences. The text is also well written. The presented data fully supports the conclusions drawn.

The study clarifies and unifies several previous works with quantitative details that exceed in quality previously published data. However, I am lacking a little the conceptual novelty. The observations reveal a highly redundant system that is able to separate chromosomes with remarkable robustness. Each of the three major and previously known pathways contribute to chromosome alignment and segregation with CLS-2 mediated pushing having a particularly prominent role. It is very exciting that in principle each of these pathways are conserved and play roles, to varied extent in diverse animal species. On the other hand, the specific constellation appears to be rather specific to *C. elegans* oocytes, and is somewhat difficult to extrapolate to other species due to the rather specialized holocentric kinetochore architecture as well as the very small size of the spindle.

Taken together, this is a very carefully carried out and informative study, which in principle could be published as is. However, in its current form I am not sure it has the sufficient level of novelty that would justify publication in a generalist journal, such as Nature Communications. For example, if the authors additionally assessed the robustness of the system, for instance by mild perturbations of microtubule dynamics, and thereby could reveal how the redundant mechanisms then provide protection against such perturbations that could potentially be very exciting.

Dear Reviewers,

We are grateful to both the Editor and Reviewers for their time, helpful comments, and rapid turn-around on the review of our paper. We appreciate this opportunity to resubmit a revised manuscript and have addressed all feasible experiments suggested by the Reviewers. We edited the manuscript (all textual changes highlighted in yellow in the revised version of our manuscript) and the figures and have included a point-by-point description of our response to each Reviewer comment below (blue text).

REVIEWER COMMENTS

Reviewer #1 (Remarks to the Author):

In the present manuscript, Pitayu-Nugroho, Aubry et al from Julien Dumont's group address the role of different kinetochore components and microtubule associated proteins during oocyte meiosis in the holocentric nematode *C. elegans*.

There has been a gap in our knowledge around the role of kinetochore components during *C. elegans* oocyte meiosis and the current work establishes a new analysis pipeline that allows the tracking of individual homologous chromosomes during Meiosis I. This level of resolution provides new information that was not attainable in previous analyses. Above all, it provides unequivocal insight into the ploidy state of the oocyte, which is key because previous studies focused on the spatial separation of two chromosome masses without a proper understanding of what that DNA content was. Therefore, the analysis can identify aneuploid oocytes that had seemingly segregated their homologs without problems. Taking advantage of this, the authors show that homologous chromosomes displaying higher orientation defects are more likely to co-segregate.

Pitayu-Nugroho, Aubry et al set to dissect the contribution of different kinetochore/chromosome proteins/complexes in chromosome orientation, congression, and correct segregation. To this end, the authors focus on 1) HIM-10, part of the NDC-80 complex; 2) SKA-1, part of the NDC-80-associated SKA complex; 3) ZWL-1, part of the RZZ complex, which recruits the motor Dynein to kinetochores; 4) the chromokinesin KLP-19; and 5) the CLASP orthologue CLS-2. They conclude that holocentric chromosome bi-orientation is not strictly required for accurate chromosome segregation. Instead, they suggest that initial kinetochore-localized CLS-2-dependent pushing acts redundantly with Ndc80 complex-mediated pulling for accurate chromosome segregation in meiosis.

In summary, the current work provides a significant step further in our understanding of meiotic chromosome segregation in the holocentric *C. elegans*. The experiments are well performed, controlled, and the quality of the data is excellent and presented in a very clear way.

I would be very much in favor of publication after some concerns/comments (listed below) are presented to the authors. I will not need to see the manuscript again as the points I raise should be addressed at the authors' discretion.

We thank Reviewer 1 for their constructive comments and we are glad they found the data of excellent quality and our manuscript publishable as it is. We nevertheless made substantial changes to the text and figures and are hopeful they will address all their concerns and comments.

1. I am not clear what the authors mean by BHC 'module'. They have shown previously that BUB-1 targets HCP-1, which in turns recruits CLS-2. However, we don't know if that is how the complex is composed *in vivo*. Importantly, these proteins don't share 100% localisation, so I felt it was at times misleading to refer to BHC to functions attributed to either BUB-1, HCP-1, or CLS-2. In this manuscript for example, the authors show regulation of CLS-2 localisation and still refer to the results as BHC module.

We agree with Reviewer 1 that it is currently unclear if BUB-1, HCP-1 and CLS-2 form a bona fide complex *in vivo*. Although our recently published manuscript¹, demonstrated that the three proteins work in a synergistic manner to control microtubule dynamics and chromosome segregation, and that interactions between them must be intact in order to fulfil these functions, we could not formally demonstrate that they assemble as a stable complex. This is precisely why we termed this association of BUB-1, HCP-1 and CLS-2 a 'module', and not a 'complex'. All the experiments related to this module in our present manuscript were performed using a KNL-1 mutant (KNL-1^{Δ85-505}), which leads, upon depletion of endogenous KNL-1, to the delocalization of all three BHC module components (BUB-1, HCP-1 and CLS-2) from kinetochores. This is primarily why we referred to these experiments as 'BHC module perturbations'. The reason for emphasizing CLS-2 function is because we previously showed that it acts as the primary effector for controlling microtubule dynamics within the module^{1, 2}. We have modified the text to make this point clearer in the revised version of our manuscript.

2. I got slightly confused as to when the authors use *in utero* and *ex utero* imaging. Could that be further clarified?

We apologize for not being clearer. *In utero* imaging is presented in Figure 1 only. All other live imaging experiments were performed *ex utero*. We have modified Figure 1 to make this point clearer in the revised version of our manuscript.

3. While the authors mention that they 'analysed the contribution of the main chromosomal components involved in contacts with microtubules', key proteins like KLP-7 (MCAK) which are present in ring domains and in chromosome/kinetochore are not mentioned or analysed. This is somewhat surprising given previous work from the group showing that KLP-7 is key during meiosis I.

We agree with Reviewer 1 that it would have been very interesting to include in our present manuscript a characterization of KLP-7 function in chromosome segregation using our novel 4D live imaging approach. However, as previously demonstrated by us and others³⁻⁵, KLP-7 perturbations lead to very severe spindle defects with mostly apolar spindles. As the goal of our present study is to analyze how the spindle interacts with chromosomes to promote their accurate segregation, disorganized spindles preclude concluding on that specific point. Furthermore, the apolar spindle phenotype precludes determining the spindle long axis and positioning of the spindle center, which are two key elements in the quantitative approach we developed for this study. These are the primary reasons for not including a characterization of KLP-7 in our study. We nevertheless included a new set of experiments, in a 'File for Reviewers only', analyzing a KLP-7 deletion mutant filmed using our 4D live imaging

protocol. These new results further confirmed the apolar spindle phenotype, and high degree of aneuploidy, observed in most meiosis I oocytes in this mutant, compared to control oocytes filmed in the same conditions.

4. It appears that KLP-19 and ZWL-1 play a role in orientation at early stages (i.e. Prometaphase I, Figure 3a,b,d), while KNL-1 depletion affects later stages, closer to Metaphase I (Figure 2d). What seems to be an addition of the two effects is achieved in two conditions, as summarised in Figure 3d: 1) KLP-19 and ZWL-1 + HIM-10 depletion (orange line) and 2) *knl-1^{Δ85-505}* + HIM-10 depletion.

a. In KLP-19/ZWL-1/HIM-10 depletion, I presume KNL-1 is still there and likely BUB-1 and CLS-2 as well, is this correct?

This assumption by Reviewer 1 is most likely correct. However, as shown in ⁶, KNL-1 kinetochore levels are slightly reduced upon Ndc80 complex perturbation. It is thus likely to also be the case here.

b. For *knl-1^{Δ85-505}*/HIM-10 depletion, I believe KLP-19 should still be present in the ring domain, along with BUB-1 and CLS-2 (Macaisne et al). Do the authors think that KLP-19 is not able to sustain lateral attachments in this condition? Does RZZ/Dynein localise normally in the *knl-1^{Δ85-505}* mutant? I might be missing something here, but it would be worth explaining these results and what the authors interpret from them in a more detailed way.

We thank Reviewer 1 for raising these important and interesting points. KLP-19, BUB-1, HCP-1 and CLS-2 should indeed still be present in the '*knl-1^{Δ85-505}*/HIM-10 depletion' condition. Reviewer 1 is also correct when assuming that dynein localizes normally at kinetochores in the *knl-1^{Δ85-505}* mutant upon KNL-1 depletion. We included a new dataset to specifically demonstrate this later point quantitatively in a revised version of our manuscript (see Extended Data Figure 6j-l). The strong mis-orientation and mis-alignment phenotypes observed in the '*knl-1^{Δ85-505}*/HIM-10 depletion' condition therefore suggest that KLP-19 and dynein are either not able to sustain lateral attachments in this condition, or not in an efficient manner. We unfortunately do not have any way to distinguish between these two hypotheses. In either case, this suggests that KLP-19 and dynein become 'inefficient' when the BHC module is absent from kinetochores. This is indeed a very interesting point and we have added a new paragraph in the discussion section of our revised manuscript to mention it.

5. The authors use RZZ component depletion to remove kinetochore Dynein (which is clearly shown). Does non-kinetochore Dynein, which seems present in the spindle area and next to chromosomes from Metaphase, play a role in orientation and/or congression? Have the authors analysed the effect of KLP-19 and Dynein co-depletion (not KLP-19 and RZZ)?

We again agree with Reviewer 1 that it would be very interesting to characterize dynein's function outside of kinetochores. However, the primary goal of our study is to understand how chromosomal activities (MAPs and kinetochores) interact with spindle microtubules to control accurate chromosome segregation. Furthermore, as previously demonstrated by us and others^{2,7}, dynein perturbations induce a severe spindle disorganization phenotype with

abnormally long spindles with unfocused and disorganized spindle poles. This again precludes determining the spindle long axis with sufficient accuracy.

6. Figures 4a-c could be slightly and inadvertently misleading by the fact that they do not start at zero. I think the zooming is important to show the trend, but it also gives the false impression of 'proximity to zero'.

This has been corrected in the revised version of Figure 4.

7. How was the 'low' dose of nocodazole established?

The low dose of nocodazole is based on ⁸. This is a dose at which some microtubules persist around chromosomes in wild-type oocytes.

8. In Figure 4g, do the different transgenes express to similar levels?

Similar expression of the three analyzed transgenes was previously demonstrated ^{9, 10}.

9. While the discussion and model explain that SKA participates in end-on attachments (bivalent stretching) closer to Metaphase I, this is not so clear in the text, where a focus on the timescale could be emphasised.

This has now been emphasized in the main text and discussion sections of our revised manuscript.

Also, if the analysis shown in 4h was done at 100 s before anaphase onset (instead of 10 s), does the $\Delta ska1$ still have an effect? If so, it would be difficult to attribute to its kinetochore localisation.

We thank Reviewer 1 for suggesting this experiment. We have now quantified chromosome stretching in the $\Delta ska1$ mutant at 150 s before anaphase onset (to make sure all quantified chromosomes are devoid of SKA) and confirmed that chromosome length at this timing is comparable to controls at the same timepoint. Therefore, the strong effect observed in the $\Delta ska1$ mutant 10 s before anaphase onset can be confidently attributed to SKA complex loading at kinetochores. We added this important control in Extended Data Figure 4h.

10. In the subheading and text related to Figure 5, it would be more accurate to refer to CLS-2 rather than the 'BHC module', since BUB-1 and HCP-1 are not analysed, and they could be playing other roles as well.

As explained in response to Point 1 raised by this Reviewer, all the experiments related to CLS-2 in our present manuscript were in fact performed using a KNL-1 mutant (KNL-1 ^{$\Delta 85-505$}), which leads, upon depletion of endogenous KNL-1, to the delocalization of all three BHC module components (BUB-1, HCP-1 and CLS-2) from kinetochores. We currently do not have any way to specifically prevent CLS-2 localization to kinetochores (without affecting its midbivalent localization). Although we are in favor of a dominant role of CLS-2, within the BHC module, in the process we describe in our study, we cannot exclude that BUB-1 and HCP-1 are also

participating. Because we analyze a mutant in which all three proteins are delocalized from the kinetochore, we think it is more appropriate to refer to these perturbations as 'BHC module' perturbations.

11. Also related to CLS-2, it appears that *knl-1^{Δ85-505}* leads to a re-localisation of CLS-2, rather than or in addition to a reduction in intensity. In fact, the midbivalent CLS-2 localisation is clearer in the *knl-1^{Δ85-505}* mutant. Hence, the statements in lines 391-393 seems like an over-simplification of what's happening.

We respectfully disagree with Reviewer 1. Our quantifications in Figure 5a show that, although CLS-2 can be seen at the mid-bivalent in both KNL-1^{FL} (wild-type) and KNL-1^{Δ85-505}-expressing oocytes, CLS-2 level is higher at the mid-bivalent in KNL-1^{FL}-expressing oocytes. Also, as requested by Reviewer 2, we have included representative live cell imaging stills of the two conditions in the revised Figure 5 (b, d), which clearly show CLS-2 at the mid-bivalent even in KNL-1^{FL} (wild-type)-expressing oocytes. The seemingly clearer mid-bivalent CLS-2 signal in KNL-1^{Δ85-505}-expressing oocytes could be due to the lack of kinetochore and spindle signals in this condition.

Reviewer #2 (Remarks to the Author):

Correct chromosome segregation in mitosis and meiosis is the basis for continued cell proliferation and gamete production, respectively. During cell division, accurate segregation of the chromosomes depends on the interaction between spindle microtubules and kinetochores to ensure timely chromosome congression and orientation, and alignment at the metaphase plate. While the molecular players at kinetochores and their function during mitotic cell divisions are well understood, our knowledge of how faithful chromosome segregation during meiosis is achieved is incomplete. As accurate chromosome segregation in female gametes is critical to avoid the generation of aneuploid eggs, progress in understanding how this is achieved, is seminal. Because of the difficulties associated with studying meiosis in mammalian oocytes, model systems such as *D. melanogaster* and *C. elegans*, as used here, are important research tools for the analysis of meiosis. In this manuscript, Pitayu-Nugroho and colleagues use a high-resolution 4D live imaging methodology to track the six bivalent chromosomes and meiotic spindle of *C. elegans* oocytes to analyse chromosome segregation in *C. elegans* oocytes. Through the systematic depletion of kinetochore proteins, the authors reveal the roles of each kinetochore component, exposing critical differences in the mechanisms of accurate chromosome segregation between organisms with holocentric and monocentric chromosomes. They show that initial lateral interactions between kinetochores and microtubules promote timely chromosome orientation and congression, while end-on attachments were observed to be particularly critical for chromosome alignment but not separation. Interestingly, in holocentric *C. elegans* oocytes there does not seem to be a requirement for bi-orientation before chromosome segregation to achieve correct segregation. At the molecular level, the NDC80 complex, the RZZ complex and the *C. elegans* KIF4A homologue *kfp-19* all contribute to meiotic chromosome segregation, but even when disturbed in combination do not give the same severe chromosome segregation phenotype as depletion of KNL1. However, combining the loss of NDC80 and the KNL1-associated BHC module, containing BUB1, as well as CENP-F and CLASP homologues, results in severe perturbation of chromosome segregation, similar to

loss of KNL1. This leads the authors to conclude that pushing forces, generated by the BHC complex, and pulling forces, created by the NDC80 complex, are required together to achieve accurate chromosome segregation in *C. elegans* oocytes.

The manuscript advances our understanding of meiotic chromosome segregation, and on the whole, the study is carefully conducted and the results are convincing. There are some issues, though, that should be addressed before publication.

We thank Reviewer 2 for his appreciation of our study and we are glad they found our results convincing.

Major points:

- The authors suggest a dominant role for the BHC complex in generating pushing forces required for correct chromosome segregation. However, at no point in the study, is the BHC complex directly analysed; in all experiments where the goal is to study the function of the BHC complex, the version of KNL1 which lacks all MELT motifs is employed. It should be demonstrated that the observed defect is indeed due to the loss of the BHC complex and not some other defect connected to the loss of this region in KNL1.

We agree with Reviewer 2 that this is an important issue. However, we previously demonstrated that none of the BHC module components can be depleted in *C. elegans* oocytes without inducing a complete disorganization of the meiotic spindle⁶. This phenotype then precludes analyzing specifically chromosome segregation and the role of kinetochores in mediating the interaction with spindle microtubules in this process (see also response to Reviewer 1). We also recently showed that even point mutations or domain-specific deletions in BHC components that preclude BHC module integrity, all phenocopy full BHC loss-of-function (i.e., formation of apolar disorganized meiotic spindles)¹. Furthermore, despite all our attempts, we never succeeded in generating a *bub-1* separation-of-function allele that would specifically prevent kinetochore recruitment of the BHC module, while keeping BHC module integrity and without inducing the full loss-of-function BHC phenotype. Thus, we currently do not have any mean of preventing kinetochore recruitment of the BHC module apart from using the KNL-1 mutant lacking all MELT repeats. However, we are confident that the observed defects, when this mutant is combined with Ndc80 complex perturbations, cannot be attributed to the loss of the MELT region of KNL-1 directly, as this mutant alone does not display any phenotype (see Extended Data Figure 5).

- When KNL-1 is replaced by KNL-1 Δ 85-505 without MELT motifs, is it clear that BUB1, HCP-1/2 and CLS-2 are not at kinetochores anymore? This needs to be demonstrated.

This has been demonstrated in our recently published manuscript¹.

- The interpretation of the data obtained upon the depletion of different kinetochore complexes depends critically on the successful depletion of the proteins of interest. It therefore needs to be demonstrated that all the RNAis are working successfully to the same extent, and differences in phenotype severity are not just due to differences in depletion efficiency.

We thank Reviewer 2 for raising this important point. All the double-stranded RNA (dsRNA) used in this study have previously been validated with the exception of the *nud-2*-targeting dsRNA. This is why we used this dsRNA only to verify the lack of kinetochore dynein (Extended Data Figure 2b), and not for functional analyses where we used the *nud-2* deletion strain instead. We have now added a supplementary table with reference to the validation of every dsRNA used in our study.

This is particularly pertinent since the authors point out that for the *C. elegans* NDC80 complex varying degrees of depletion obtained by different research groups may have resulted in differential phenotypes observed by these groups.

We do not think that the differential phenotypes associated with Ndc80-complex perturbations, and reported in the literature, result from variations in the degree of depletion. Instead, and as discussed in our manuscript, we think that they primarily stem from the different live imaging approaches used by different groups, which can induce photodamage and lead to artificial aggravation of this phenotype.

- Can the authors comment on or provide evidence against the independency of kinetochore proteins? For example, does depleting components of Ndc80 complex affect KNL1 localisation as it does mammalian cells?

As suggested by Reviewer 2, some level of interdependency between kinetochore components also exists in the *C. elegans* oocyte. This has been extensively studied in ⁶ and was taken into account for the interpretation of results from our experiments. Figure 1a comprehensively summarizes all interdependencies between kinetochore components in the *C. elegans* oocyte. The revised version of this figure displays an, initially omitted, thin arrow to depict the fact that depleting NDC-80 indeed leads to a slight reduction in KNL-1 levels at kinetochores, as shown in ⁶. However, this modest decrease in KNL-1 levels is clearly insufficient to phenocopy KNL-1 depletion itself or even to induce any mis-segregation event. We thank reviewer 2 for pointing out to this omission.

Minor points:

- The order of the figures compared to the flow of text is slightly confusing due to the regular return to additional panels of figure 3. Some figure rearrangement may be required.

This has been edited in the revised version of Figure 3. We have now split the data originally presented in Figure 3 to create a new Extended Data Figure 3.

- Can the authors comment on the minor segregation defects observed for the double KLP-19 and ZWL1 depletion? Data suggests that chromosomes reach a point comparable to controls but a mild increase in mis-segregation is observed.

The mild level of mis-segregation observed in this condition corresponds to a single oocyte with 4 lagging chromosomes and 1 co-segregating bivalent. Although, we do not report a similar case in control unperturbed oocytes in the present study, this level of aneuploidy can also be observed stochastically in this condition. We and others made previous report of such cases in previously published studies^{3, 6}.

- Could representative live cell imaging stills be shown for Figure 5a and b, so the reader can evaluate the differences in mitotic progression?

These have been added in the revised version of Figure 5 (see panel b and d).

Reviewer #3 (Remarks to the Author):

The manuscript by Pitayu-Nugroho and coworkers provides a systematic analysis of chromosome segregation in *C. elegans* oocytes.

Firstly, they develop a live-imaging assay to image chromosome segregation from its beginning to the end in oocytes. For this, they acquire fast 3D stacks of samples with chromatin (H2B-mCherry) and microtubules (tubulin-EGFP) labeled. They then analyze data using a semi-automated workflow. These analyses provide a highly quantitative characterization of the entire process with impressive resolution. The authors can precisely measure, in a large number of oocytes, chromosome orientation, chromosome alignment, axial compaction. In addition, measuring the angles of bivalents turned out to be an exceptionally informative parameter.

This assay is then used throughout the manuscript and combined with systematic genetic perturbations of kinetochore components. The authors show that KLP-19 and dynein-/dynactin-mediated lateral interactions promoted timely chromosome orientation and congression in prometaphase. They further show the requirement for the Ndc80 complex for generating pulling forces through end-on attachments, mediating chromosome alignment in metaphase. Interestingly, neither of these two complexes is strictly required for accurate chromosome segregation. Chromosome segregation is instead primarily mediated by the BHC module (composed of BUB-1, CENP-F and CLASP), which facilitates central spindle assembly and generates pushing forces to separate kinetochores in anaphase.

The manuscript is beautifully illustrated with figures that visualize imaging data as well as quantitative analyses in a very intuitive way. This is further helped by excellent schemes, which make it easy to follow even complex perturbations and their consequences. The text is also well written. The presented data fully supports the conclusions drawn.

The study clarifies and unifies several previous works with quantitative details that exceed in quality previously published data. However, I am lacking a little the conceptual novelty. The observations reveal a highly redundant system that is able to separate chromosomes with remarkable robustness. Each of the three major and previously known pathways contribute to chromosome alignment and segregation with CLS-2 mediated pushing having a particularly prominent role. It is very exciting that in principle each of these pathways are conserved and play roles, to varied extent in diverse animal species. On the other hand, the specific constellation appears to be rather specific to *C. elegans* oocytes, and is somewhat difficult to extrapolate to other species due to the rather specialized holocentric kinetochore architecture as well as the very small size of the spindle.

Taken together, this is a very carefully carried out and informative study, which in principle could be published as is.

However, in its current form I am not sure it has the sufficient level of novelty that would justify publication in a generalist journal, such as Nature Communications. For example, if the authors additionally assessed the robustness of the system, for instance by mild perturbations of microtubule dynamics, and thereby could reveal how the redundant

mechanisms then provide protection against such perturbations that could potentially be very exciting.

We thank Reviewer 3 for their helpful suggestion and we are delighted they found our study carefully carried and informative. We thought that the demonstration of a novel mechanism controlling accurate meiotic chromosome segregation, that uniquely does not rely on Ndc80 complex-mediated chromosome bi-orientation, is already quite novel. However, to address Reviewer 3 comment, we have performed a new set of experiments where we perturbed microtubule dynamics by deleting the Kinesin-13 family member KLP-7. As predicted by this reviewer, and despite the severe spindle defects observed in *klp-7*-deleted oocytes, most chromosomes (67%) segregated accurately in this condition. This result highlights, as suggested by Reviewer 3, the high level of protection against aneuploidy provided by the redundant mechanisms that control accurate chromosome segregation in the *C. elegans* oocyte. Although, we think these new results are particularly interesting, we also feel that they would require further investigation before being presented in a publishable manner. We will in fact pursue this analysis, which will be the topic of a future study. This is why we have decided to present these data in a 'File for Reviewers Only' rather than to include them in our current, already quite dense, study. We are thus hopeful Reviewer 3 will now be convinced by the level of novelty of our findings and find the manuscript suitable for publication.

1. Macaisne, N. *et al.* Synergistic stabilization of microtubules by BUB-1, HCP-1, and CLS-2 controls microtubule pausing and meiotic spindle assembly. *Elife* **12** (2023).
2. Laband, K. *et al.* Chromosome segregation occurs by microtubule pushing in oocytes. *Nat Commun* **8**, 1499 (2017).
3. Gigant, E. *et al.* Inhibition of ectopic microtubule assembly by the kinesin-13 KLP-7/MCAK prevents chromosome segregation and cytokinesis defects in oocytes. *Development* (2017).
4. Connolly, A.A., Sugioka, K., Chuang, C.H., Lowry, J.B. & Bowerman, B. KLP-7 acts through the Ndc80 complex to limit pole number in *C. elegans* oocyte meiotic spindle assembly. *J Cell Biol* **210**, 917-932 (2015).
5. Han, X., Adames, K., Sykes, E.M. & Srayko, M. The KLP-7 Residue S546 Is a Putative Aurora Kinase Site Required for Microtubule Regulation at the Centrosome in *C. elegans*. *PLoS One* **10**, e0132593 (2015).
6. Dumont, J., Oegema, K. & Desai, A. A kinetochore-independent mechanism drives anaphase chromosome separation during acentrosomal meiosis. *Nature cell biology* **12**, 894-901 (2010).
7. van der Voet, M. *et al.* NuMA-related LIN-5, ASPM-1, calmodulin and dynein promote meiotic spindle rotation independently of cortical LIN-5/GPR/Galpha. *Nature cell biology* **11**, 269-277 (2009).
8. Hyman, A.A. & White, J.G. Determination of cell division axes in the early embryogenesis of *Caenorhabditis elegans*. *The Journal of cell biology* **105**, 2123-2135 (1987).

9. Cheerambathur, D.K., Gassmann, R., Cook, B., Oegema, K. & Desai, A. Crosstalk Between Microtubule Attachment Complexes Ensures Accurate Chromosome Segregation. *Science* **342**, 1239-1242 (2013).
10. Cheerambathur, D.K. *et al.* Dephosphorylation of the Ndc80 Tail Stabilizes Kinetochore-Microtubule Attachments via the Ska Complex. *Dev Cell* **41**, 424-437 e424 (2017).

REVIEWERS' COMMENTS

Reviewer #2 (Remarks to the Author):

The authors have addressed all my concerns, and I am happy to support publication.

Reviewer #3 (Remarks to the Author):

As an outsider to the *C. elegans* community I apparently did not fully appreciate the novelty and importance of the presented work. The other reviews and the detailed rebuttal letter, however, convinced me about the relevance of the findings for the specific field, and therefore I am happy to support the publication of the revised version of the manuscript. I also appreciate that the authors performed additional experiments to address the specific point I have raised -- I hope this may become part of a future study.